# Water-insoluble organic carbon in $PM_{2.5}$ over China: light-absorbing properties, potential sources, radiative forcing effects and possible light-absorbing continuum

Yangzhi Mo[1, 2], Jun Li[*, 1, 2], Guangcai Zhong[1, 2], Sanyuan Zhu[1, 2], Shizhen Zhao[1, 2], Jiao Tang[1, 2], Hongxing Jiang[3], Zhineng Cheng[1, 2], Chongguo Tian[4], Yingjun Chen[3], Gan Zhang[1, 2]

[1] State Key Laboratory of Organic Geochemistry and Guangdong province Key Laboratory of Environmental Protection and Guangdong-Hong Kong-Macao Joint Laboratory for Environmental Pollution and Control, Guangzhou Institute of Geochemistry, Chinese Academy of Science, Guangzhou 510640, China

[2] CAS Center for Excellence in Deep Earth Science, Guangzhou, 510640, China

[3] Shanghai Key Laboratory of Atmospheric Particle Pollution and Prevention (LAP3), Department of Environmental Science and Engineering, Fudan University, Shanghai 200438, China

[4] Key Laboratory of Coastal Environmental Processes and Ecological Remediation, Yantai Institute of Coastal Zone Research, Chinese Academy of Sciences, Yantai, 264003, China

[*]Corresponding Authors: Dr. Jun Li

E-mail: junli@gig.ac.cn; Tel: +86-20-85291508; Fax: +86-20-85290706

## Abstract

Water-insoluble carbon (WIOC) constitutes a substantial portion of organic carbon (OC) and contributes significantly to light absorption by brown carbon (BrC), playing pivotal roles in climate forcing and human health. China as hotspots regions with high level of OC and BrC, information regarding the sources and light-absorbing properties of WIOC on national scale remains scarce. Here, we investigated the light-absorbing properties and sources of WIOC in ten representative urban cities across China. On average, WIOC accounted for 33.4 ± 7.66% and 40.5 ± 9.73% of the concentrations and light-absorbing efficiency at 365 nm ($Abs_{365}$) of extractable OC (EX-OC, comprising relatively hydrophobic OC [WIOC and humic-like substances: HULIS-C], and hydrophilic OC [non-humic-like substances: non-HULIS-C]). The mass absorption efficiency of WIOC at 365 nm ($MAE_{365}$) was (1.59 ± 0.55 $m^2/gC$) comparable to that of HULIS (1.54 ± 0.57 $m^2/gC$) but significantly higher than non-HULIS (0.71 ± 0.28 $m^2/gC$), indicating

that hydrophobic OC possesses a stronger light-absorbing capacity than hydrophilic OC. Biomass burning (31.0%) and coal combustion (31.1%) were the dominant sources of WIOC, with coal combustion sources exhibited the strongest light-absorbing capacity. Moreover, employing the simple forcing efficiency ($SFE_{300-700nm}$) method, we observed that WIOC exhibited the highest $SFE_{300-700nm}$ (6.57 ± 5.37 W/g) among the EX-OC fractions. The radiative forcing of EX-OC was predominantly contributed by hydrophobic OC (WIOC: 39.4 ± 15.5% and HULIS: 39.5 ± 12.1%). Considering the aromaticity, sources, and atmospheric processes of different carbonaceous components, we propose a light-absorbing carbonaceous continuum, revealing that components enriched with fossil sources tend to possess stronger light-absorbing capacity, higher aromatic levels, increased molecular weights, and greater recalcitrance in the atmosphere. Reducing fossil fuel emissions emerges as an effective means of mitigating both gaseous ($CO_2$) and particulate light-absorbing carbonaceous warming components.

## Highlights

- WIOC contributed significantly to both concentrations and the light absorption efficiency of extractable organic carbon.
- WIOC primarily originated from biomass burning and coal combustion in China.
- WIOC exhibited the highest radiative forcing among the extractable organic fractions.
- Carbonaceous components that are more enriched with fossil sources tend to exhibit stronger light-absorbing capacity, higher aromatic levels and molecular weight, and enhanced recalcitrance

## 1. Introduction

Organic carbon (OC) constitutes a substantial fraction (20 to 90%) of carbonaceous aerosols, playing an important role in human health, air quality and climate change (Jimenez et al., 2009; Zhang et al., 2007). Recent studies have shown that specific organic compounds could efficiently absorb radiation in near-ultraviolet (UV) and visible spectral regions, exhibiting a strong wavelength dependence (Laskin et al., 2015; Andreae and Gelencser, 2006). Due to its brownish or yellowish visible appearance, the light-absorbing OC is term as brown carbon (BrC) (Sun et al., 2007; Saleh, 2020). Currently, model studies showed that the BrC account for ~20 to 40% of the light absorption of total carbonaceous aerosols

absorption globally, hence, BrC has the potential to counteract the cooling effects of OC, introducing considerable uncertainty into climate models (Bahadur et al., 2012; Feng et al., 2013; Saleh et al., 2015). Moreover, BrC may contribute to the generation of reactive oxygen species (ROS) in ambient aerosols, posing potential adverse effects on human health (Verma et al., 2012; Wang et al., 2023). To comprehensively understand and address the climate and health impacts of BrC, there is a critical need for thorough investigations into the sources and light-absorbing properties of OC.

The common technique to investigate OC light-absorbing properties is to use the spectrophotometry to determine the light absorption of OC extracted by water or solvents with different polarity (Chen and Bond, 2010; Liu et al., 2013; Hecobian et al., 2010; Chen et al., 2017). This approach can effectively eliminate confounding influence of insoluble light-absorbing particle (e.g., black carbon and mineral dust). Crucially, it also allows for the sequential study of the sources and light-absorbing properties of BrC within different OC components characterized by distinct polarities (Xie et al., 2017; Chen et al., 2016; Huang et al., 2020; Verma et al., 2012; Mo et al., 2017; Jiang et al., 2020b). According to the water solubility, the OC can be classified as water-soluble OC (WSOC) and water-insoluble OC (WIOC). While extensive investigations have been conducted on the sources, light-absorbing properties, and atmospheric processes of WSOC over the past decades (Bosch et al., 2014; Dasari et al., 2019; Mo et al., 2021; Wozniak et al., 2014; Wang et al., 2020). WIOC, constituting a substantial portion of OC (up to ~80%), has received relatively limited attention (Xie et al., 2017; Huang et al., 2020; Mihara and Mochida, 2011; Sciare et al., 2011). Recent studies indicate that both the light-absorbing capacity and light absorption by WIOC are higher than WSOC (Chen and Bond, 2010; Liu et al., 2013; Cheng et al., 2016), attributed to the enrichment of more potent light-absorbing BrC chromophores in WIOC (Lin et al., 2018; Huang et al., 2020; Zhang et al., 2013). For examples, Zhang et al. (2013) reported that the light absorption by methanol-extracted OC in Los Angeles was approximately 3 and 21 times higher than that by WSOC. Additionally, certain BrC chromophores, such as polycyclic aromatic hydrocarbons (PAHs) and their derivatives, present in the WIOC fraction pose a high risk of lung cancer (Aquilina and Harrison, 2023). Further, methanol-extracted OC has been identified as a significant contributor to the oxidative potential of aerosols (Gao et al., 2020; Verma et al., 2012). Indeed, WIOC in aerosols has been shown to induce cytotoxic, genotoxic, oxidative, and inflammatory effects in MRC-5 human lung epithelial cells (Velali et al., 2016). Field observations further indicate that WIOC exhibits greater recalcitrance than WSOC during long-range transport processes, resulting in a longer lifetime for WIOC compared to WSOC (Wozniak et al., 2012; Fellman et al., 2015;

Kirillova et al., 2014). The WIOC as relatively longer lifetime OC component with higher light-absorbing capacity and toxicity, therefore, comprehensive understanding of the sources and light-absorbing properties of WIOC is imperative.

China as the hotpot regions of OC, the columnar mass concentration of BrC in China (4.4 to 92 mg/m$^2$) is much higher than those in Europe and U.S.A (~5 mg/m$^2$) (Arola et al., 2011; Zhang et al., 2017). While the sources and light-absorbing properties of WSOC have been extensively investigated in China (Huang et al., 2020; Jiang et al., 2020b; Wang et al., 2023; Cheng et al., 2016; Yan et al., 2017; Mo et al., 2021), corresponding information on WIOC remains limited, especially on a national scale. In this study, we selected ten representative Chinese cities with urbanization rates ranging from 37.8% to 88.0% to represent the regions with different developed levels. The primary objectives of this study are to explore the spatiotemporal variations in concentrations, light absorption properties, sources, and radiative effects of WIOC across these urban areas in China. Additionally, we integrate and make a comparation of light-absorbing properties data (mass absorption efficiency [MAE] and absorption Ångström exponent [AAE]) of OC with different polarities (hydrophobic WSOC isolated by solid-phase extraction referred to as humic-like substances [HULIS] and the hydrophilic WSOC referred to as non-HULIS), and BC from previous studies (Mo et al., 2021; Mo et al., 2024). Finally, we propose a continuum concept of light-absorbing carbonaceous aerosols linked to aromaticity, sources, and atmospheric processes. This study provides insights into light-absorbing properties and sources of WIOC, contributing essential knowledge for a comprehensive understanding the role of WIOC in climate forcing and developing strategies to mitigate its climate impact.

# 2. Materials and Methods

## 2.1 Sampling

$PM_{2.5}$ samples were collected across four seasons in ten cities in China. These cities included four with central heating systems (Beijing, Xinxiang, Lanzhou, and Taiyuan) and six without central heating (Shanghai, Nanjing, Chengdu, Guiyang, Wuhan, and Guangzhou), as shown in Figure 1. All the filter samples were collected on pre-combusted (450°C, 6h) quartz-fiber filter (Pall, England), use a high-volume sampler at a flow rate of ~ 1000L/min. Detailed information about the sampling methods can be found in

our previous study (Mo et al., 2021; Mo et al., 2024). In brief, each sampling campaign spanned approximately 30 days for fall, winter, spring, and summer, respectively. Subsequently, a 20 mm diameter sample was excised from each filter during every season, and these were amalgamated into a single sample, with the exception of Guiyang, where only fall and winter samples were available. A total of 38 pooled samples were utilized in subsequent experiments. For each location, one pooled sample was obtained for each season, thus providing analytical results representing seasonal averages.

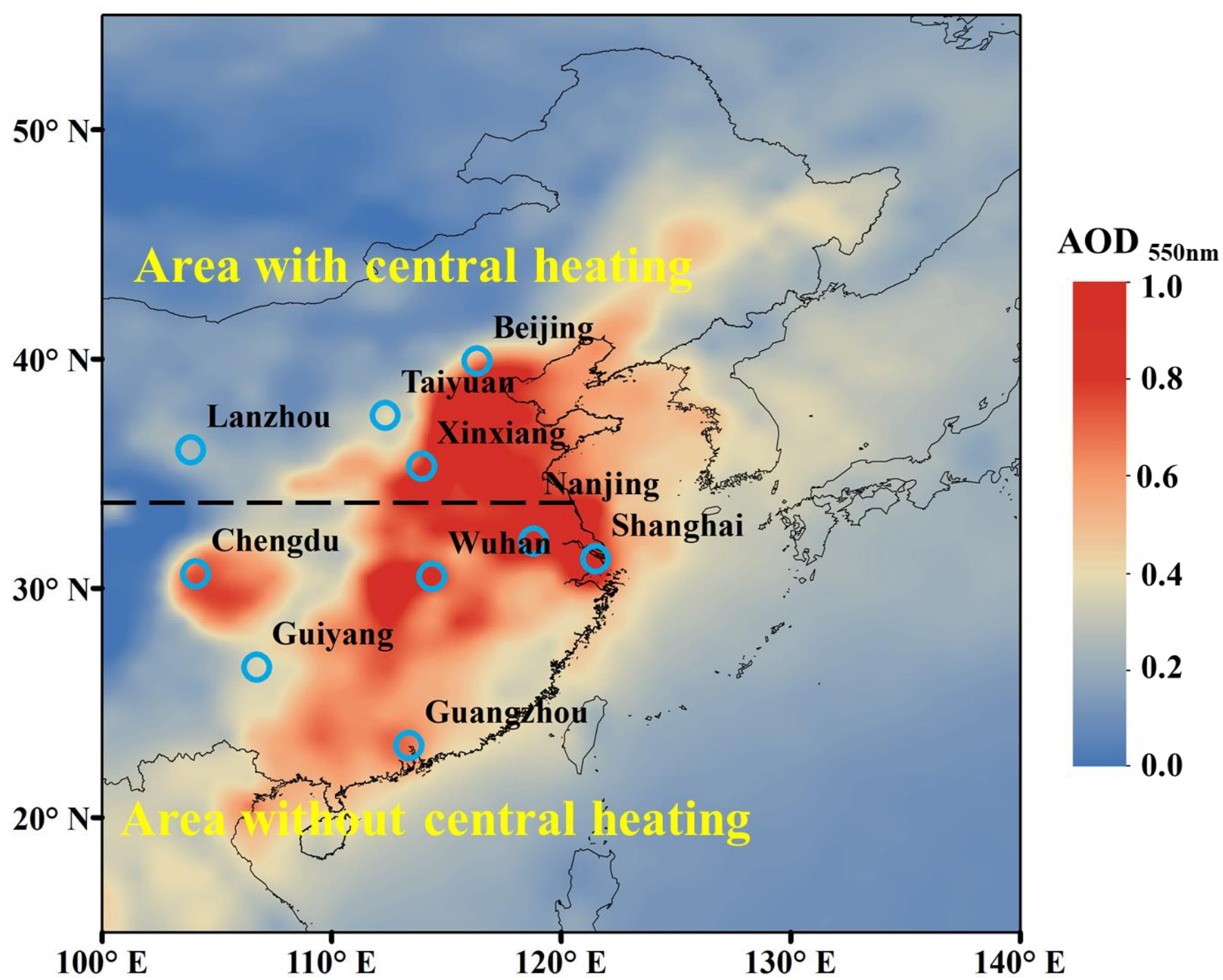

**Figure 1.** The average aerosol optical depth (AOD) at 550 nm retrieved from satellite (Terra/MODIS) observations over East Asia during October 2013 to August 2014. The locations of ten Chinese cities are shown in the map. Beijing, Xinxiang, Lanzhou and Taiyuan located in the areas with central heating in cold seasons (fall and winter). Shanghai, Nanjing, Chengdu, Guiyang, Wuhan and Guangzhou located in the areas without central heating.

## 2.2 Chemical analysis

For water-soluble inorganic ions analysis, the filters were ultrasonically extracted with ultrapure water (18.2 MΩ cm) in a polypropylene vial for 30 min. Extracts were filtered through polytetrafluoroethylene

(PTFE) syringe filters (Jinteng Ltd., Tianjin, China) of 0.22 μm pore size to remove particles and filter debris. Seven water soluble inorganic ions ($Na^+$, $NH_4^+$, $K^+$, $Mg^{2+}$, $Cl^-$, $SO_4^{2-}$, and $NO_3^-$) were determined by ion-chromatography (761 Compact IC, Metrohm, Switzerland). The detection limit was below 0.05 mg/L for all ions.

For the solvent extraction, the water-soluble organic carbon (WSOC) in the pooled sample was extracted with 100 mL ultrapure water (18.2 MΩ, Sartorius) under ultrasonication (30 min × 3 times). Following previous studies demonstrating that most water-insoluble organic carbon (WIOC) can be extracted in methanol (> 90%) (Chen and Bond, 2010; Chen et al., 2017; Cheng et al., 2016), the same sample underwent drip drying, and the WIOC was re-extracted methanol (OCEANPAK, HPLC-Grade, 30 min × 3 times) using the same procedure. Both the methanol and water extracts were filtered through a 0.22 μm PTFE membrane to remove insoluble particles. Both the methanol and water extracts were filtered through a 0.22 μm PTFE membrane to remove insoluble particles. The WSOC further separated into a relatively hydrophobic (humic-like substance, HULIS) and hydrophilic (non-HULIS) fraction following a solid-phase extraction (SPE) as described in previous studies (Lin et al., 2010; Fan et al., 2012).

The WSOC and HULIS-C content was determined using a TOC analyzer equipped with a nondispersive infrared (NDIR) detector (Shimadzu TOC-VCPH, Japan). The non-HULIS-C was estimated by the difference between WSOC and HULIS-C (non-HULIS-C = WSOC - HULIS-C). For WIOC measurement, 40 mL methanol extracts were evaporated to dryness under a nitrogen stream and re-dissolved with 1.0 mL methanol. A 20 μL aliquot of extracts was slowly spiked onto a 1.5 $cm^2$ prebaked quartz filter. After methanol evaporation, carbon on the quartz filter was quantified with an OC/EC analyzer (Sunset Laboratory Inc). The relative standard deviation was within 3%. Based on extractable OC (EX-OC) polarity, the EX-OC was separated into WIOC, HULIS-C, and non-HULIS-C.

## 2.3 Light absorption spectra measurement

The absorption spectra of sloven extracted fractions were recorded from 200 to 800 nm relative to ultrapure water by a UV-visible spectrophotometer (UV-4802, Unico, China). The light absorption coefficient was calculated according to following equation (Hecobian et al., 2010; Kirillova et al., 2014):

$$\mathrm{Abs}_\lambda = (\mathrm{A}_\lambda - \mathrm{A}_{700})\frac{\mathrm{V}_l}{\mathrm{V}_a \times l} \times \ln(10) \qquad (1)$$

where $Abs_\lambda$ is the light absorption coefficient ($Mm^{-1}$), $V_l$ is the volume of solvent for extraction (ml), $V_a$ is the volume of sampled air ($m^3$), $l$ is the optical path length (in this case, 0.01 m), and $A_\lambda$ is the absorption of the solution at a given wavelength. The average light absorption between 695 and 705 nm ($A_{700}$) was used to account for baseline drift during analysis. The mass absorption coefficient (MAE, $m^2$/gC) of sloven extracted OC fractions at wavelength of λ can be calculated as:

$$MAE_\lambda = \frac{Abs_\lambda}{C_i} \qquad (2)$$

Where $C_i$ is the corresponding concentration of WIOC, HULIS-C and non-HULIS-C in the air (μgC/$m^3$).

The wavelength dependence of different OC fraction can be investigated by fitting the absorption Ångström exponent (AAE) by the following relation:

$$Abs_\lambda = K \times \lambda^{-AAE} \qquad (3)$$

The AAE is calculated by a linear regression of ln ($Abs_\lambda$) on ln(λ) within the range 330-400 nm for the avoidance of interference by non-organic species (e.g., $NO_3^-$). The ratio of light absorption at 250 and 365 nm (E2/E3), which is negatively correlated with aromaticity and molecular weight of organics was also calculated (Peuravuori and Pihlaja, 1997; Baduel et al., 2010).

## 2.4 Positive matrix factorization (PMF) source apportionment

We applied U.S EPA PMF 5.0 model to qualitatively and quantitatively identify sources of WIOC and $Abs_{365, WIOC}$ in this study. The principle and detailed process of this model could be found in Paterson (1999) and EPA 5.0 Fundamentals & User Guide. PMF model is a commonly used mathematical approach for the apportionment of $PM_{2.5}$ sources abase on the characteristic chemical compositions or fingerprints in each source. The model decomposes the concentrations of the chemical species of samples (X) into sets of contributions (G), factor profiles (F), and residuals (E):

$$X = G \times F + E \qquad (4)$$

During the model calculation, factor contributions and profiles were derived by minimizing the objective function Q in PMF model:

$$Q = \sum_{i=1}^{m}\sum_{j=1}^{n}\left(\frac{E_{ij}}{\sigma_{ij}}\right)^2 \qquad (5)$$

where $E_{ij}$ is the residual of each sample, and $\sigma_{ij}$ is the uncertainty in the jth species for the sample i.

The measurement uncertainties were used for the error estimates of the measured concentrations. Data values below the method detection limit (MDL) were substituted with MDL/2. Missing data values were substituted with median concentrations. If the concentration is less than or equal to the MDL, the corresponding uncertainty (Unc) is 5/6 MDL. Otherwise, the uncertainty is calculated following equation:

$$\mathrm{Unc} = \sqrt{(\text{error fraction} \times \text{concentration})^2 + (0.5 \times \mathrm{MDL})^2} \quad (6)$$

We performed 100 random runs and retained the runs that produced minimum Q values for 3 to 10 factors in base runs, five factors were obtained as the optimal solution as the source profiles in this study (Figure S1).

## 2.5 Radiative effect calculation

The "simple forcing efficiency" (SFE, W/g) proposed by Bond and Bergstrom (2006) was used to estimate the potential direct radiative effects caused by light-absorbing OC. The SFE was originally used to represent the normalization of the particle mass (Chylek and Wong, 1995). Here, we focused on the light absorption effect of OC without the scattering effect. A wavelength-dependent SFE of light-absorbing OC as follows (Chen and Bond, 2010) :

$$\frac{\mathrm{dSFE_{abs}}}{\mathrm{d}\lambda} = \mathrm{D}\frac{\mathrm{dS}(\lambda)}{\mathrm{d}\lambda}\tau_{\mathrm{atm}}^2(1-\mathrm{F_c}) \times 2\alpha_s \times \mathrm{MAC_i} \quad (7)$$

where S and $\tau_{atm}$ refer to solar irradiance and atmospheric transmission, respectively, with both being from ASTM G173–03 reference spectra (W/m$^2$). D is the daytime fraction (0.5), Fc is the cloud fraction (0.6), and αs is the surface albedo (0.19 for Earth average). MACi is mass absorption cross section of sloven extracted OC (e.g., WIOC, HULIS and non-HULIS). Note that MAC refers to the particulate absorption per mass, while MAE is derived from absorption of the aqueous extracts. MAC can be compared with MAE only after considering the particulate effect (Sun et al., 2007) (as described in Text S1). And then, the fraction of solar radiation absorbed by OC component with different polarity relative to total EX-OC is calculated as:

$$f_{OC_i/EX_OC} = \frac{\sum_{\lambda=300}^{700} SFE_{OC_i}(\lambda) \times C_i \times \left(\frac{OA}{OC}\right)}{\sum\sum_{\lambda=300}^{700} SFE_{OC_i}(\lambda) \times C_i \times \left(\frac{OA}{OC}\right)} \quad (8)$$

Here, the integrated SFE is the sum of the SFE from 300 to 700 nm; $C_i$ is the corresponding

concentration of WIOC, HULIS-C and non-HULIS-C in the air (μgC/$m^3$). The OA/OC ratios are 1.51, 1.91, 2.30 for WIOC, HULIS and non-HULIS, respectively (Kiss et al., 2002).

# 3. Results and Discussion

## 3.1 Spatiotemporal variations of concentration and light-absorbing properties of WIOC

In this study, water-insoluble organic carbon (WIOC) is defined as the residual OC re-extracted by methanol after water extraction, representing the OC only soluble in methanol. We define the OC extracted by water from the aerosol filter sample as WSOC, the WSOC is further separated into hydrophobic fraction (HULIS-C) and hydrophilic fraction (non-HULIS-C). The combined sum of WSOC and WIOC is defined as extractable OC (EX-OC). Figures 2 shows the spatial variation of concentration and $Abs_{365}$ of separated EX-OC fractions across ten Chinese. The concentrations of WIOC ranged from 1.45 to 12.95 μgC/$m^3$, with an average of 3.64 ± 2.53 μgC/$m^3$ among the 10 cities (Figure 2a). Specifically, the areas with central heating exhibited significantly higher average WIOC concentrations compared to areas without central heating (4.79 ± 3.39 μgC/$m^3$ vs. 2.81 ± 1.16 μgC/$m^3$, $p < 0.01$), likely attributed to coal and biofuel combustion for domestic/central heating during the cooler period (Wang et al., 2023; Wang et al., 2020). Despite substantial spatial variation in WIOC concentration, its contribution to EX-OC remained consistent at 33.4 ± 7.6%, showing no significant spatial or temporal variations. Furthermore, the fractional carbon mass contributions of WIOC (33.4 ± 7.6%), HULIS-C (35.2 ± 5.8%), and non-HULIS-C (31.4 ± 5.2%) to EX-OC were comparable (Figure 2 and Table S1).

Consistent with spatial variation in WIOC concentration, the $Abs_{365}$ of WIOC ($Abs_{365,\ WIOC}$) serving as a proxy for BrC were significantly higher in areas with central heating compared to those without central heating (10.1 ± 10.3 $Mm^{-1}$ vs. 4.41 ± 2.68 $Mm^{-1}$, $p < 0.01$). Notably, the light absorbing contribution of WIOC (40.5 ± 9.73%) to EX-OC exceeded its corresponding carbon mass contribution (33.4 ± 7.55%). Actually, the light absorbing contribution of EX-OC are largely contributed by relatively hydrophobic OC components: the WIOC (40.5 ± 9.73%) and HULIS (41.6 ± 7.28%). In contrast, the non-HULIS fraction, being the most polar, contributed only 17.5 ± 5.02% to $Abs_{365,\ EX\text{-}OC}$ (Table S1). This suggests that the majority of light-absorbing organic compounds were enriched in the WIOC and HULIS fractions. Therefore, the mean mass absorption efficiency (MAE) spectra of WIOC and HULIS, representing the light-absorbing

capacity per unit carbon mass, were higher than those of non-HULIS (Figure 1c).

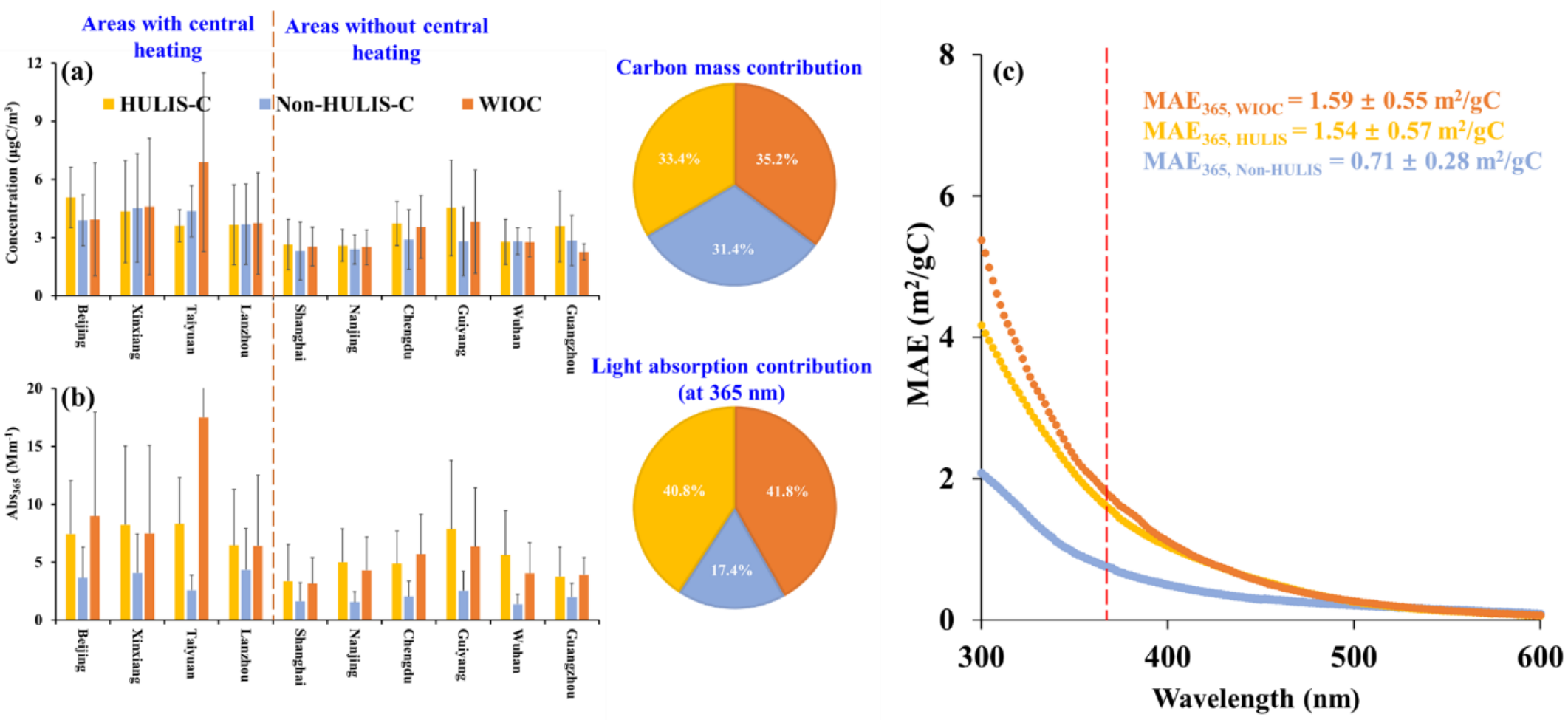

**Figure 2.** The spatial variation of concentration and light absorption of extractable OC from ten Chinese cities. (a) The spatial variations of concentration of WIOC, HULIS-C and non-HULIS-C in $PM_{2.5}$ from ten Chinese cities; (b) The spatial variations of light absorption coefficients of WIOC, HULIS-C and non-HULIS-C at 365 nm ($Abs_{365}$) in $PM_{2.5}$ from ten Chinese cities; The pie charts in the left of panel (a) and (b) represent the carbon mass contribution and light absorption contribution, respectively. (c) The mean of mass absorption efficient (MAE) of WIOC, HULIS and non-HULIS from 300 nm to 600 nm; The red dash line represents the $MAE_{365}$.

The MAE at 365 nm ($MAE_{365}$) is commonly used to reflect the light-absorbing capacity of solvent extracted-BrC. Among the extractable OC components, the $MAE_{365}$ of WIOC is the highest, with average of 1.59 ± 0.55 $m^2$/gC. This value is comparable to the WIOC in Xi'an (1.5 ± 0.5 $m^2$/gC) and Beijing (1.5 ± 0.4 $m^2$/gC) (Huang et al., 2020), but ~5 times higher than values reported in Nagoya, Japan (0.2 to 0.4 $m^2$/gC) (Chen et al., 2016). The $MAE_{365}$ of WIOC is comparable to HULIS (1.54 ± 0.57 $m^2$/gC), however, higher than the non-HULIS as relatively polar water-soluble fraction (0.71 ± 0.28 $m^2$/gC). This discrepancy is likely attributed to the non-HULIS fraction mainly comprising highly oxidized organic matter lacking long aromatic conjugated systems (Chen et al., 2017; Chen et al., 2016). While WIOC stands out as the most light-absorbing OC component, its MAE at 550 nm (0.14 ± 0.09 m2/gC) remains an order of magnitude lower than that of amorphous tar ball BrC (~3.6 to 4.1 $m^2$/g) and unextractable "dark BrC" (~1.2 $m^2$/g) determined by transmission electron microscopy (Alexander et al., 2008; Chakrabarty et al., 2023),

indicating the light-absorbing capacity of the extractable BrC is relatively weakly.

The $MAE_{365}$ of WIOC exhibited significant seasonal variation, with higher values in cold seasons ($1.74 \pm 0.64$ $m^2/gC$, fall and winter) than in warm seasons ($1.48 \pm 0.46$ $m^2/gC$, spring and summer, Figure 3a). This variation is likely linked to changes in sources and atmospheric processes influencing the light-absorbing compounds within the WIOC fraction. During cold seasons, large usage of coal combustion and BB for central/domestic heating may elevate the emission of the WIOC with high $MAE_{365}$ (Tang et al., 2020; Song et al., 2019), consequently enhancing the overall $MAE_{365}$ of WIOC. Conversely, stronger photobleaching effects and lower emissions from coal combustion and BB during warm seasons may contribute to a decrease of $MAE_{365}$ of WIOC (Saleh et al., 2013; Wong et al., 2017). Interestingly, all extractable OC components exhibit a consistent seasonal pattern (cold > warm) in their $MAE_{365}$, indicating similar influences of sources and atmospheric processes on the light-absorbing capacity of these components irrespective of polarity.

The distinct seasonal variation of light-absorbing capacity of WIOC may be affected by the structure of light-absorbing compounds within WIOC. AAE reflects both the wavelength dependent light absorption and aromaticity of the carbonaceous aerosols, and the AAE usually negatively related with the aromaticity (Chen et al., 2017; Mo et al., 2017; Zhang et al., 2013). BC as most condensed aromatic and strongest light-absorbing carbonaceous, exhibits an AAE of ~1 (Bond, 2001; Kirchstetter et al., 2004). In the case of BrC in solvent extracts, AAE values typically vary from ~3 to 16 (Hecobian et al., 2010; Mo et al., 2021; Chen and Bond, 2010). Generally, during photobleaching aging processes, the MAE of BrC in solvent extracts tends to decrease with increasing AAE values, indicative of a reduction in aromaticity (Dasari et al., 2019). Despite the stronger radiation and lower $MAE_{365}$ of WIOC in warm seasons, AAE values did not exhibit seasonal variation ($4.59 \pm 0.52$ vs. $4.77 \pm 0.65$, $p > 0.05$, Figure 3b). This may be due to the more complex factors affecting the AAE values, which are not only affected by sources and atmospheric processes (Saleh et al., 2013; Tang et al., 2020; Dasari et al., 2019), but also by the solvents and the pH of water extracts applied in the determination (Chen et al., 2016; Mo et al., 2017; Phillips et al., 2017). However, the AAE values for WIOC ($4.69 \pm 0.59$) were comparable to those of HULIS ($4.72 \pm 0.53$) but lower than those of non-HULIS ($7.33 \pm 2.56$). This suggests a tendency for AAE to increase with the polarity of OC components, in agreement with findings from Los Angeles and Nagoya (Chen et al., 2016; Zhang et al., 2013). This also indicated that relatively hydrophobic fractions (e.g., WIOC and HULIS) contain more

aromatic light-absorbing compounds than non-HULIS.

In contrast, the seasonal variation is more pronounced for the ratio of light absorption at 250 and 365 nm (E2/E3), inversely proportional to the molecular weight (MW) and aromaticity of natural organic matter (Baduel et al., 2010; Peuravuori and Pihlaja, 1997). The E2/E3 of WIOC was lower in cold seasons compared to warm seasons (4.33 ± 0.49 vs. 4.51 ± 0.0.48, $p < 0.01$, Figure 3c), suggesting the WIOC exhibited greater conjugations and aromaticity in cold seasons. Notably, the E2/E3 ratio exhibits a stronger correlation with combustion source tracers (e.g., $K^+$ and $Cl^-$) during cold seasons ($K^+$: $r = 0.37$, $p = 0.02$ vs. $r = 0.34$, $p > 0.1$; $Cl^-$: $r = 0.56$, $p = 0.011$ vs. $r = 0.16$, $p > 0.1$) than in warm seasons, indicating that coal combustion and BB contribute to higher aromaticity of WIOC during cold seasons (Duarte et al., 2005; Fan et al., 2016). Indeed, coal combustion and BB are important sources of OC with high level aromatic compounds (e.g., PAHs). Additionally, the slower photo-degradation and volatilizations of aromatic compounds in lower temperature also enhanced the aromatic level of WIOC in cold seasons (Samburova et al., 2007; Zhang et al., 2020b). Similar to AAE, the E2/E3 of extractable OC components exhibited a consistent trend of increasing with the polarity of OC (WIOC: 4.41 ± 0.49 < HULIS: 4.93 ± 0.50 < non-HULIS: 7.00 ± 0.42, $p < 0.01$), suggesting that less polar organics likely have higher aromaticity and higher MW. However, in contrast to AAE, all E2/E3 ratios of the EX-OC components exhibited the same seasonal variation (cold > warm, Figure 3c). This implies that the E2/E3 ratio, calculated using two wavelengths, maybe more effective than the AAE calculated using multiple wavelengths when reflecting changes in the structure of organic components. The seasonal variation in the structure and light-absorbing properties of WIOC is largely influenced by sources, we further explore the source of WIOC, as discussed below.

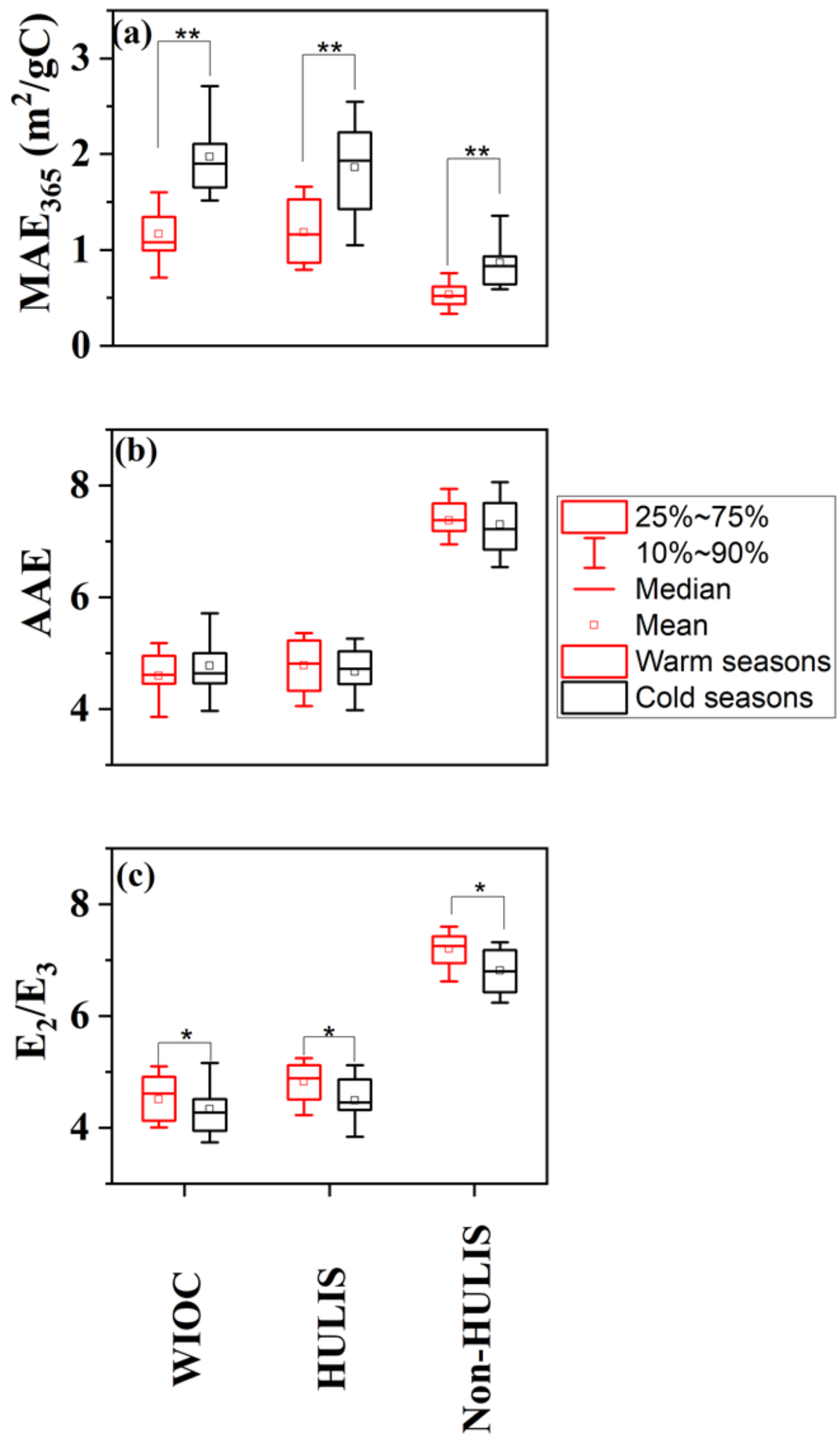

**Figure 3.** The seasonal variations of (a) AAE, (b) E2/EC and (c) $MAE_{365}$ for the WIOC, HULIS and WSOC. "*" indicates the difference at $p < 0.05$ level; "**" indicates the difference at $p < 0.01$ level

## 3.2 Coal combustion and biomass burning dominate the sources of WIOC, with coal combustion sources exhibited the strongest light-absorbing capacity

In order to better understand the source of WIOC and BrC in WIOC fraction, correlation of WIOC and $Abs_{365, WIOC}$ to water-soluble ions were investigated. The WIOC correlated well with $Abas_{365, WIOC}$ ($r = 0.97$,

$p < 0.01$), suggesting the WIOC had similar sources and formation processes of the light-absorbing compounds in WIOC. However, as listed in Table S2, the correlations of WIOC and $Abs_{365, WIOC}$ with water-soluble ions differed notably between warm and cold seasons. During cold seasons, both WIOC and $Abs_{365, WIOC}$ exhibited good relationship with most of the water-soluble ions (Table S2). Conversely, in warm seasons, WIOC showed poor correlation with most water-soluble ions, except for $NH_4^+$ ($r = 0.51$, $p < 0.05$). Compared with the WIOC, the $Abs_{365, WIOC}$ correlated well with most water-soluble ions ($r = 0.63$, 0.53, 0.51 and 0.59 for $Cl^-$, $NO_3^-$, $SO_4^{2-}$, and $NH_4^+$, respectively, $p < 0.05$, Table S2), except the $K^+$ ($r = 0.25$, $p > 0.05$), in warm seasons. This suggests that the difference in sources and formation processes in WIOC and light-absorbing compounds in warm seasons.

To better quantify the seasonal variation of the sources of WIOC and BrC in WIOC fraction, we quantified the sources for both WIOC concentrations and light absorption of WIOC ($Abs_{365, WIOC}$) using the PMF receptor model in this study. The model identified five factors with uncertainties below 12%, and their profiles are presented in Figure S1. Factor 1 exhibited a high $Cl^-$ loading (57.0%), a typical tracer for BB and coal combustion. However, $K^+$ as typical tracer for BB, the loading of $K^+$ (11.7%) was relatively low in Factor 1. Further, Factor 1 displayed a ratio of $Abs_{365, WIOC}$ to WIOC (2.46 $m^2/gC$) comparable to the $MAE_{365}$ of WIOC from coal combustion (Tang et al., 2020). Consequently, Factor 1 was classified as a source related to coal combustion. Factor 2 was characterized by the highest loading of $K^+$ (45.8%) and HULIS-C (44.1%), thus, this factor was identified as BB. Factor 3 exhibited enrichment in $SO_4^{2-}$ (46.0%) and non-HULIS-C (18.2%), both recognized as key components in atmospheric aging processes (Du et al., 2014). Notably, the ratio of $Abs_{365, WIOC}$ to WIOC in Factor 3 was the lowest (0.55 $m^2/gC$) among the identified factors. This observation suggests a probable loss of light-absorbing capacity during aging/bleaching processes. Thus, Factor 3 is interpreted as the source associated with aging processes. Factor 4 is related to high loading of $NO_3^-$ (56.2%) and $NH_4^+$ (34.8%). Given that SOA formed under high $NO_x/NH_3$ conditions often exhibits high light-absorbing capacity (Xie et al., 2017; Lin et al., 2018), and considering the relatively high ratio of $Abs_{365, WIOC}$ to WIOC (2.13 $m^2/gC$) observed in Factor 4, we attribute Factor 4 to be a source related to nitrogen-induced SOA formation. Factor 5, characterized by the highest loading of $Ca^{2+}$ (65.7%). Ca is identified as tracer of fugitive dust (Han et al., 2007). The predominance of Ca in Factor5, which points to sources such as resuspended dust and soil sources. Both the predicted WIOC concentrations ($R^2 = 0.92$) and $Abs_{365, WIOC}$ ($R^2 = 0.91$, Figure S2) correlated well with the corresponding measured values, confirming the reliability of PMF solution.

Figure 4a shows the annual average contributions of the identified sources to WIOC resolved by PMF model. The primary sources of WIOC were combustion sources, with coal combustion and BB averagely account for 31.1% and 31.0% of the WIOC, respectively. In contrast, sources related to aging processes and nitrogen-induced secondary formation accounted for 18.2% and 5.2% of the WIOC, respectively. That may be due to these two secondary sources are more enriched in water-soluble components (HULIS-C + non-HULIS-C). Actually, although the uncertainties of sources contribution of HULIS-C and non-HULIS-C resolved by PMF model may be high, the sources of aging processes (HULIS-C:10.1% and non-HULIS-C: 18.3%) and nitrogen-related secondary formation (HULIS-C: 20.2% and non-HULIS-C: 21.6%) were relatively enriched in WSOC fraction. Notably, the sources contribution from different sources exhibited distinctive seasonal variation (Figure 3b). In winter, coal combustion dominated the sources of WIOC (48.4%), likely associated with the increased usage of coal for central/domestic heating. In contrast, during the summer, when both temperature and solar radiation intensity rise, the contribution from coal combustion decreased to 2.8%, while the contributions from aging processes and BB increased to 39.3% and 41.3%, respectively. In spring, a significant fraction of WIOC was associated with dust/soil, reaching up to 28.8%, likely due to the higher wind speed leading to an abundance of dust particles in the city.

Figure 4c shows the contributions of identified sources to the $Abs_{365, WIOC}$. Generally, coal combustion (46.5%) and BB (30.0%) dominate $Abs_{365, WIOC}$, while other sources just contribute 23.8% (aging processes: 6.1%, nitrogen-related secondary formation: 6.7%, and dust/soil: 10.8%) of $Abs_{365, WIOC}$. Annually, even though the mass contributions of coal combustion and BB to WIOC are comparable, coal combustion is the largest contributor to $Abs_{365, WIOC}$, surpassing BB. This difference is likely because coal-derived WIOC has a stronger light-absorbing capacity than BB. The seasonal variation of sources contribution of $Abs_{365, WIOC}$ is similar to that of mass contribution. Coal combustion is the dominant contributor to $Abs_{365, WIOC}$ in winter (62.5%), but its contribution diminishes in other seasons, suggesting that the enhanced light absorption of WIOC in winter is driven by coal combustion. In contrast to previous studies reported that a main contribution of secondary sources to light absorption of BrC in summer (Du et al., 2014; Yan et al., 2017), we found that the light absorption of WIOC is primarily from BB. This discrepancy may be attributed to the fact that secondary light-absorbing compounds are mostly present in water-soluble components and are less prevalent in water-insoluble components. In spring, the contribution of dust/soil to $Abs_{365, WIOC}$ reaches up to 26.6%, likely due to the presence of humic substances with strong light-absorbing capacity in dust/soil

(Andreae and Gelencser, 2006).

The source of BrC significantly influences its light absorption capacity. For WIOC, the contribution from aging processes shows a negative correlation with the $MAE_{365}$ of WIOC ($r = -0.61$, $p < 0.01$, Figure S3a), indicating that the chromophores in WIOC were bleached during aging processes. Conversely, the contribution from coal combustion is positively correlated with the $MAE_{365}$ of WIOC ($r = 0.72$, $p < 0.01$), suggesting that light-absorbing compounds derived from coal combustion have a strong light-absorbing capacity and enhance the overall $MAE_{365}$ of WIOC (Figure S3a). Moreover, the contribution from coal combustion is also significantly positively correlated with the light absorption contribution of WIOC to EX-OC ($Abs_{365,\ WIOC}/Abs_{365,\ EX\text{-}OC}$, $r = 0.46$, $p < 0.01$, Figure S3b). This implies that the strong light-absorbing compounds emitted from coal combustion tend to be water-insoluble. It is noteworthy that, based on carbon isotopes ($\delta^{13}C$ and $\Delta^{14}C$), coal combustion is identified as the major source of strong light-absorbing components in the water-soluble fraction in China (Mo et al., 2021; Mo et al., 2024). Consequently, a significant correlation between the coal combustion contribution and $Abs_{365,\ EX\text{-}OC}$ is observed ($r = 0.84$, $p < 0.01$, Figure S3c). Overall, coal combustion is the dominant source of both WIOC and WSOC with strong light-absorbing capacity in China, which enhances the overall color of EX-OC.

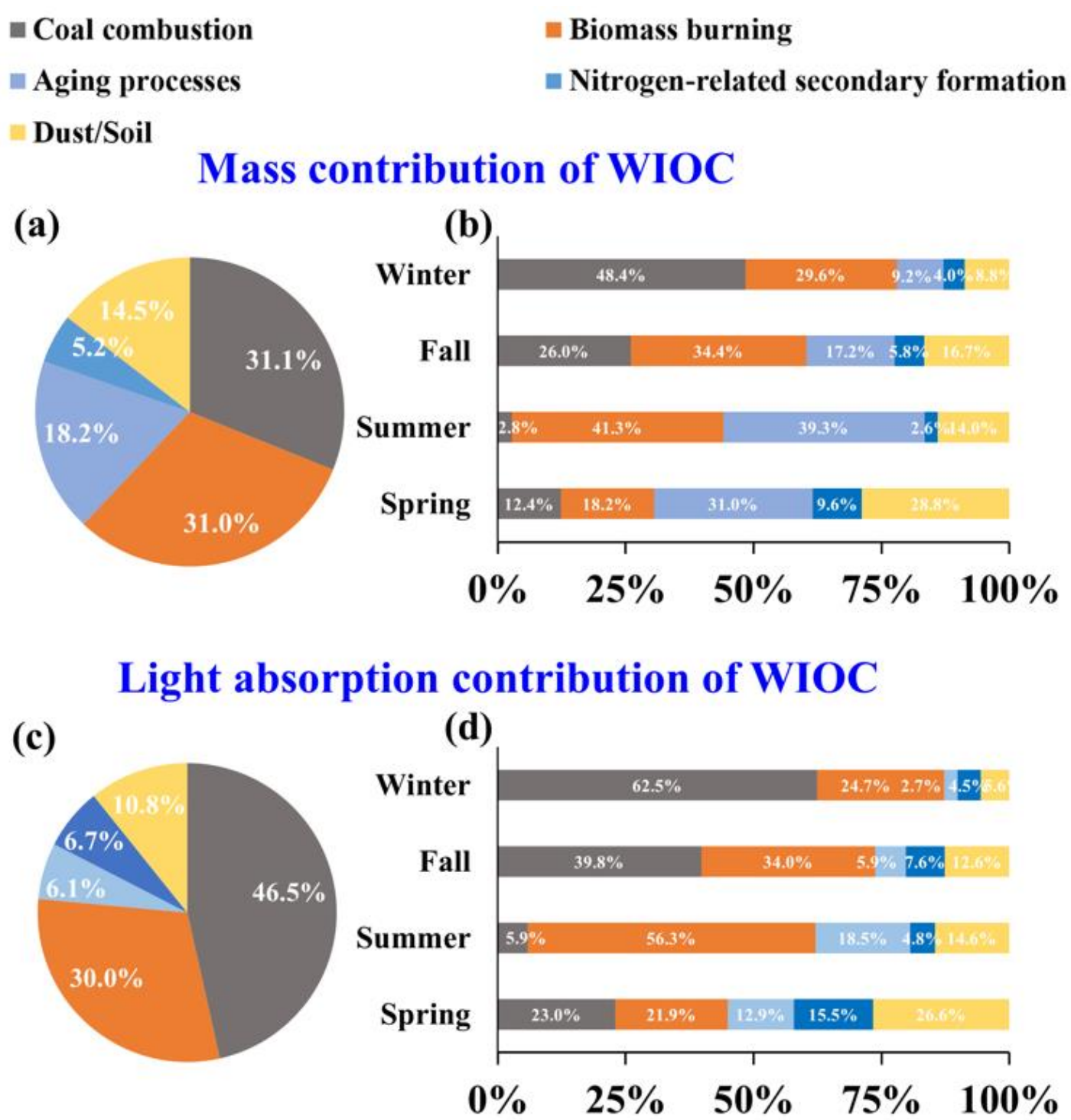

**Figure 4.** (a) Annual and (b) seasonal sources apportionments result of WIOC mass concentration. (c) Annual and (d) seasonal sources apportionments of WIOC light absorption at 365 nm.

## 3.3 Radiative forcing of WIOC

The potential radiative of WIOC was estimated by a "simple forcing efficiency" (SFE) method, as described in section 2.5 (Bond and Bergstrom, 2006; Chylek and Wong, 1995). The wavelength-dependent absorption SFE from 300 to 700 nm for WIOC, HULIS and non-HULIS are shown in Figure 5a. The integrated mean SFE from 300 to 700 nm ($SFE_{300-700}$) is highest for WIOC (6.57 ± 5.37 W/g), followed by HULIS (4.39 ± 1.79 W/g) and non-HULIS (1.23 ± 1.03 W/g). This order is consistent with the $MAE_{365}$ of these three fractions (Figure 1c). Comparing the SFE values with previous reports in Chinese cities, the values for WIOC and HULIS fall within the reported range (Hong Kong: 4.40 W/g, Tianjin: 6.30 ± 2.30 W/g, Xi'an: 3.51 ± 2.36 W/g, all for WSOC)(Deng et al., 2022; Li et al., 2023; Zhang et al., 2020a), but lower than that in Kanpur, India (19.2 W/g, for WSOC)(Choudhary et al., 2021). It's important to note that the SFE values presented here are calculated from bulk light absorbance measurements of the extracts, which tend to be lower than corresponding values from filter-based optical transmission measurements (Li et al., 2020).

The radiative effect of light-absorbing OC is generally related to its atmospheric concentration. In equation (8), the concentrations of WIOC, HULIS, and non-HULIS were further taken into account and used to estimate their relative contributions to the solar radiation absorbed by EX-OC (Figure 5b). The fraction of radiative forcing by WIOC (39.4 ± 15.5%) was almost equal to that of HULIS (39.5 ± 12.1%), but much higher than non-HULIS (21.1 ± 10.2%). This result suggests that the radiative forcing of EX-OC is dominantly contributed by the relatively hydrophobic OC fractions, making them efficient radiative-forcing agents. In contrast, consistent with previous studies, the radiative effects of oxidized OC fractions are relatively limited (Tian et al., 2023). Overall, the radiative forcing of different components of OC is highly inhomogeneous, likely associated with their sources and atmospheric processes.

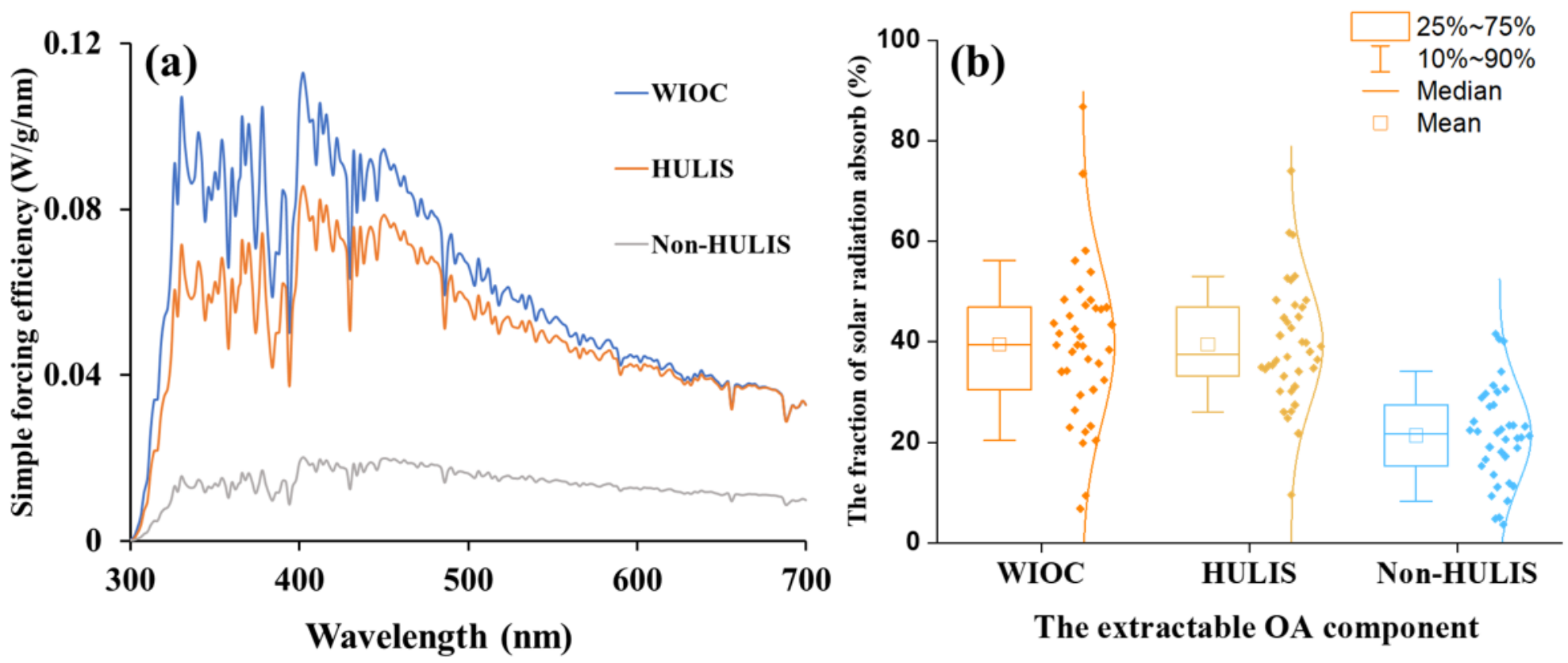

**Figure 5.** (a) The average simple forcing efficiency (SFE) of the WIOC, HULIS and WSOC from 300 to 700 nm; (b) The fraction of solar radiation absorbed by WIOC, HULIS and non-HULIS relative to total extractable OC. The relative fraction of non-HULIS is calculated by the difference between WSOC and HULIS.

## 3.4 A possible continuum of light-absorbing carbonaceous components related to the aromaticity, sources and atmospheric processes

The light-absorbing carbonaceous aerosols has conventionally been classified into BC and BrC. Following the classification framework introduced by Saleh (2020), we mapped the BC and BrC in an

AAE-log$MAE_{405}$ space. Within the space, BrC can be further categorized into the following: very weakly (VW), weakly (W), moderately (M), and strongly (S) light-absorbing BrC. In this framework, hydrophobic organic carbon (OC), represented by WIOC and HULIS, falls into the M-BrC area. On the other hand, the relatively hydrophilic OC (e.g., non-HULIS) is skewed more toward the W-BrC area (Figure 6a). It is important to note that WIOC in this study refers to OC that is insoluble in water but soluble in methanol. Thus, WIOC, HULIS, and non-HULIS are considered as extractable OC, implying that all solvent-extractable OC falls within the W- and M-BrC categories. It should be emphasized that S/Dark-BrC, characterized by light-absorbing properties similar to BC, is typically unextractable, as demonstrated in previous studies (Chakrabarty et al., 2023; Corbin et al., 2019). Similarly, BC is traditionally considered unextractable and exhibits the strongest light-absorbing capacity among the various carbonaceous components. The gradual enhancement of the light-absorbing capacity within the carbonaceous components is intricately linked to their molecular structure, specific sources, and atmospheric processes.

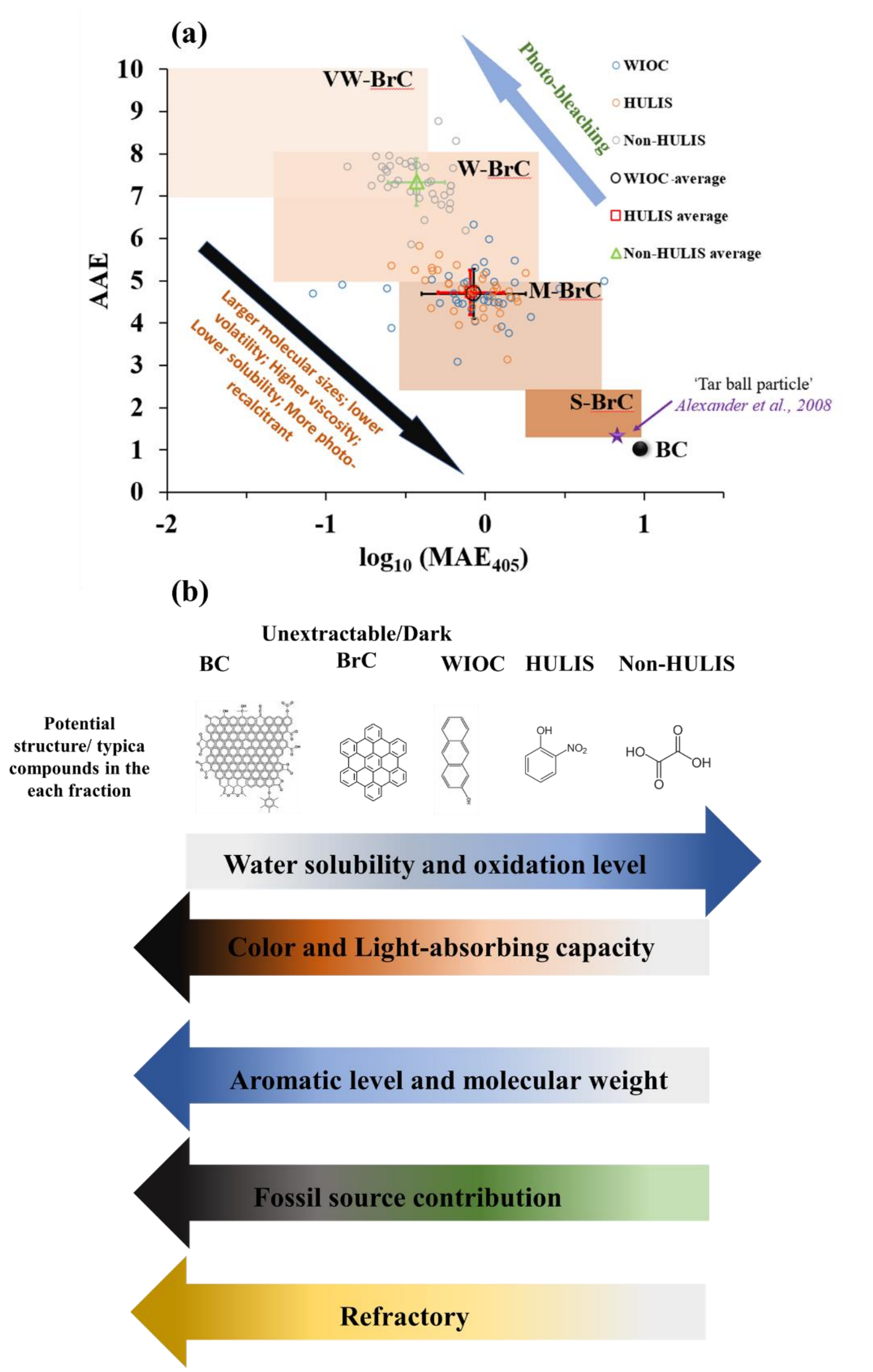

**Figure 6.** (a) The light-absorbing properties of WIOC, HULIS and Non-HULIS mapped in the in the log10($MAE_{405\ nm}$) – AAE space following the approach of Saleh (2020). Brown-shaded areas indicate "very weakly (VW)", "weakly (W)", "moderately (M)", and "strongly (S)" light-absorbing BrC classes. The star symbol marks the upper limit of individual "tar ball particles" inferred from the electron energy loss Spectro microscopy (Alexander et al., 2008).The AAE is calculated for the wavelength range 330 to 400 nm in this study. (b) The continuum of light-absorbing capacity, water solubility, aromaticity, source and refractory of different carbonaceous components. Arrows indicate the direction of increase.

In Figure 6b, we proposed a possible light-absorbing continuum of carbonaceous aerosols. The light-absorbing properties of carbonaceous aerosols are largely dependent on their molecular structure, the aromatic molecules have been shown to be the most important fractions relevant to the light-absorbing properties (Andreae and Gelencser, 2006; Bond et al., 2013; Laskin et al., 2015). Generally, the light-absorbing capacity increased with the aromatic level/fractions. For instance, most of BC and dark-BrC as the unextractable and strong light-absorbing components, are enriched with carbon-rich aromatic molecules (Corbin et al., 2019; El Hajj et al., 2021; Saleh, 2020). BC is composed of large polycyclic aromatics with graphitic-like structures (Pöschl, 2005). According to the chemical and physical properties, BC can be further subdivided into soot-BC and char-BC (Han et al., 2010; Masiello, 2004). Due to higher the $sp^2$-bond carbon and aromatic level of soot-BC, the light-absorbing capacity of soot-BC is higher than char-BC (Andreae and Gelencser, 2006; Corbin et al., 2019; Schnaiter et al., 2003). Dark-BrC, also known as tar balls, is also unextractable light-absorbing carbonaceous component, which exhibits light-absorbing properties similar to BC. The dark-BrC could be considered as incipient BC, characterized by lower molecular weight and aromatic levels compared to mature BC (El Hajj et al., 2021; Saleh et al., 2018). For the extractable hydrophobic OC fraction (e.g., WIOC and HULIS), the aromatic compounds, including PAHs, nitrophenol, and O-/N-aromatics, are the major light-absorbing components (Laskin et al., 2015). Notably, both the molecular weight and aromatic level of aromatic compounds in the extractable hydrophobic OC fraction are commonly lower than those in dark-BrC (Chakrabarty et al., 2023; Corbin et al., 2019). Thus, both the WIOC and HULIS located in the M-BrC space as shown in the Figure 6a. In this study, optical parameters (E2/E3 and AAE) do not reveal a significant difference in molecular weight and aromatic levels between HULIS and WIOC. This discrepancy may be attributed to the limited reliability and accuracy of these optical parameters in reflecting molecular weight and aromaticity. However, employing more robust analytical technologies, such as ultra-resolution mass spectrometry and benzene-poly-carboxylic acids tracers (Sun et al., 2021; Tang et al., 2020), studies has demonstrated higher aromatic fraction and aromaticity for WIOC compared to HULIS (Huang et al., 2020; Sun et al., 2021; Tang et al., 2020). Particularly, polycyclic aromatics are identified as key fractions determining both the light absorptivity and wavelength dependence of WIOC from biomass burning source samples (Sun et al., 2021). Compared with hydrophobic OC (e.g., WIOC and HULIS), the hydrophilic OC (e.g., non-HULIS) exhibits much lower molecular weight and aromaticity. This is evidenced by the much lower E2/E3 and AAE values of hydrophobic OC than the hydrophilic OC (Figures 2b and c). Overall, the light-absorbing capacity (or

color) of carbonaceous components follows the order: soot-BC > char-BC > dark-BrC > WIOC > HULIS > non-HULIS. The light-absorbing capacity and molecular weight of carbonaceous components increase with their aromaticity, while water solubility/polarity decreases with increasing aromaticity.

The molecular structure of carbonaceous components is highly related to specific sources. The BC as carbonaceous with highest aromatic level, is exclusively emitted from incomplete combustion of BB and fossil fuel (Bond et al., 2013). The content of soot-/char-BC from distinct primary emission sources significantly varies with fuel type and combustion conditions (Cai et al., 2023; Han et al., 2021). Soot-BC, formed in high-temperature combustion conditions and with high aromatic contents in the fuel, is more prevalent in fossil fuel combustion processes (e.g., coal and gasoline) than in BB (Han et al., 2021; El Hajj et al., 2021). Thus, soot-BC is predominantly contributed by fossil fuel combustion, while char-BC is the dominant subgroup in BB (Cai et al., 2023; Han et al., 2010). Remarkably, despite the subgroup distinctions within BC, as the strongest light-absorbing carbonaceous component, over 70% of BC in the ambient aerosols from major city clusters in China is attributed to fossil fuel combustion (Jiang et al., 2020a). Regarding the BrC, in addition to sharing primary emission sources with BC, it can also be formed secondarily through complex chemical reactions (Laskin et al., 2015). Dark-BrC as strongest light-absorbing OC, predominantly derived from incomplete combustion of BB and fossil fuels. Laboratory experiments and field observations consistently show that dark-BrC is more abundant in BB plumes (Chakrabarty et al., 2023; Mathai et al., 2023). However, it is noteworthy that controlled-combustion experiments report fossil fuel-derived dark-BrC may exhibit a stronger light-absorbing capacity than BB-derived dark-BrC (Cheng et al., 2019; Yu et al., 2021). Concerning the WIOC, our PMF-based sources apportionment results cannot distinctly differentiate between fossil and non-fossil sources for WIOC. However, radiocarbon isotope ($\Delta^{14}C$), a more robust source apportionment method, has demonstrated that, despite variations in sampling locations and seasons, fossil sources are more enriched in WIOC than WSOC in ambient aerosols from South/East Asia and the U.S.A. (Dasari et al., 2019; Kirillova et al., 2014; Wozniak et al., 2012; Kirillova et al., 2013). In the water-soluble fractions, the HULIS as relatively hydrophobic WSOC, our previous study using radiocarbon isotope ($\Delta^{14}C$) have shown that the HULIS across ten Chinese cities exhibit a higher fossil contribution than hydrophilic WSOC (e.g., non-HULIS) (48.9 ± 9.0 % vs. 30.3 ± 13.9 %, $p < 0.01$) (Mo et al., 2021; Mo et al., 2024). By correlating the light-absorbing capacity with the variation in the sources of different carbonaceous components, as discussed above, we observe

that more strongly absorbing carbonaceous components tend to be more enriched with fossil sources.

Upon emission or generation into the atmosphere, the light-absorbing properties of carbonaceous aerosols undergo dynamic changes significantly influenced by atmospheric processes (Dasari et al., 2019; Laskin et al., 2015). The response of different carbonaceous components to atmospheric processes varies extensively. The BC is refractory carbonaceous component in the aerosols, which is recalcitrant to the chemical oxidation. Although laboratory studies reported that the BC is possible oxidized and release water-soluble component under specific conditions (Decesari et al., 2002). But BC is often cored with OC in the ambient aerosols, making somewhat shielded towards oxidants (Bond et al., 2013). Similarly, dark-BrC exhibits considerable resistance to sunlight-driven photochemical bleaching, resulting in the persistence of light-absorbing organic aerosols in the atmosphere (Chakrabarty et al., 2023). This resistance is likely associated with the high viscosity of dark-BrC, limiting surface and bulk reaction rates. Consequently, unextractable light-absorbing components (BC + dark-BrC) not only display strong light absorptivity but also persist longer in the atmosphere. For the WIOC, based on the PMF model results, we found that the WIOC was enriched with primary emissions sources (e.g., coal combustion and BB) than WSOC. This indicated the WIOC is more recalcitrant than WSOC. Indeed, employing dual carbon isotopes ($\delta^{13}C$- $\Delta^{14}C$), studies found that WIOC is not only enriched with fossil sources, but also exhibit greater persistence and relatively longer lifetimes compared to WSOC components present in ambient aerosols (Kirillova et al., 2014; Kirillova et al., 2013; Wozniak et al., 2012). Similar to WIOC, fossil components in HULIS are more resistant and less susceptible to oxidative photobleaching, contributing to their relatively high light-absorbing capacity compared to non-HULIS components (Mo et al., 2024). Generally, chromophores in the aqueous phase experience rapid photo-bleaching, while those in the viscous organic phase undergo slower rates of photo-degradation (Klodt et al., 2022). The recalcitrant properties of WIOC and HULIS may stem from the tendency of these hydrophobic OC components to partition into the viscous organic phase, potentially rendering them more photo-recalcitrant. By linking the light-absorbing capacity to the refractory of different carbonaceous component as discussed above, the strongly light-absorbing carbonaceous components tend to be more recalcitrant in the atmospheres (Figure 6b).

Taken together, we propose a continuum for light-absorbing carbonaceous aerosols, taking into account factors such as aromaticity, molecular weight, sources, polarity and atmospheric processes (Figures 6a and b). The light-absorbing capacity of carbonaceous components exhibit following orders: soot-BC > char-

BC > dark-BrC > WIOC > HULIS > non-HULIS. This hierarchy indicates that the light-absorbing capacity of carbonaceous components increases with aromaticity, molecular weight, fossil sources contribution, and refractoriness. Conversely, as the polarity/oxidized level of carbonaceous components increases, their light absorbing-capacity weakens. These findings suggest that fossil fuel combustion tends to generate relatively long-term and strongly light-absorbing carbonaceous components. In contrast, light-absorbing carbonaceous components derived from biomass burning are prone to photo-degradation, transforming into colorless carbon with high polarity and cloud condensation activity.

## 4. Conclusions

In this study, we investigated the light-absorbing properties and sources of WIOC in ten representative urban cities across China. We found that WIOC averagely accounts for a substantial portion of the concentrations (33.4 ± 7.66%) and $Abs_{365}$ (40.5 ± 9.73%) of extractable OC (EX-OC). The $MAE_{365}$ of WIOC (1.59 ± 0.55 $m^2/gC$) was comparable to that of HULIS (1.54 ± 0.57 $m^2/gC$) but significantly higher than non-HULIS (0.71 ± 0.28 $m^2/gC$), suggesting the stronger light-absorbing capacity of hydrophobic OC (WIOC+HULIS) compared to hydrophilic OC (non-HULIS). The dominant sources of WIOC were biomass burning (31.0%) and coal combustion (31.1%), with coal combustion exhibiting the highest light-absorbing capacity among these sources. Moreover, utilizing the simple forcing efficiency ($SFE_{300\text{-}700nm}$) method, we found that WIOC exhibited the highest $SFE_{300\text{-}700nm}$ (6.57 ± 5.37 W/g) among the EX-OC fractions. Notably, the radiative forcing of EX-OC was predominantly attributed to hydrophobic OC (WIOC: 39.4 ± 15.5% and HULIS: 39.5 ± 12.1%). Finally, we proposed a light-absorbing carbonaceous continuum based on considerations of aromaticity, sources, and atmospheric processes of different carbonaceous components. This continuum revealed that carbonaceous components more enriched with fossil sources tend to possess stronger light-absorbing capacity, higher aromatic levels, increased molecular weights, and greater recalcitrance in the atmosphere. The implications of our study underscore the necessity of reducing fossil fuel emissions as an effective strategy for mitigating both gaseous ($CO_2$) and particulate light-absorbing carbonaceous warming components.

## Author Contributions

**Conceptualization:** Yangzhi Mo

**Funding acquisition:** Gan Zhang

**Investigation:** Yangzhi Mo, Jiao Tang, Hongxing Jiang, Zhineng Cheng, Sanyuan Zhu

**Methodology:** Yangzhi Mo

**Project Administration:** Gan Zhang, Shizhen Zhao

**Resources:** Sanyuan Zhu, Yingjun Chen, Chongguo Tian, Zhineng Cheng, Gan Zhang

**Software:** Yangzhi Mo

**Supervision:** Guangcai Zhong, Jun Li,Gan Zhang

**Validation:** Yangzhi Mo, Jun Li

**Writing – original draft:** Yangzhi Mo

**Writing – review & editing:** Yangzhi Mo, Jun Li, Gan Zhang

## Competing interests

The contact author has declared that none of the authors has any competing interests

## Acknowledgements

This study was supported by the Natural Science Foundation of China (NSFC; No. 42030715, 42192511, and 42107121), the Alliance of International Science Organizations (Grant No. ANSO-CR-KP-2021-05), the Guangdong Basic and Applied Basic Research Foundation (2021A0505020017, 2023A1515012359 and 2023B1515020067), and a scholarship for Yangzhi Mo provided by the China Scholarship Council (202204910172). The authors gratefully thank the people at all sites for sample collections and all of the individuals and groups participating in this project.

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
