# Peer review of "Water-insoluble organic carbon in PM$_{2.5}$ over China: light-absorbing properties, potential sources, radiative forcing effects and possible light-absorbing continuum"

_EGUsphere, 2024_

## Referee Comment (RC2)

In this study, different fractions of organic carbon in ambient fine particulate matter (PM$_{2.5}$) in ten Chinese cities and their optical effects were investigated. Results indicated that the optical effects of extractable organic carbon are mainly contributed by relatively hydrophobic fractions (i.e., water-insoluble organic carbon and humic-like substances). Both empirical indices and the source apportionment model indicate that aromatic compounds from primary emissions tend to exhibit a stronger light-absorbing capacity. This study can provide significant information on the chemical compositions and sources of brown carbon (BrC) for further mitigating the climate effects of PM$_{2.5}$. I recommend accepting this manuscript if the following comments could be addressed in the revised version.

General comments:

1. This study cannot represent the entire China, because only samples from urban areas were analyzed. All these samples were collected during 2013-2014, a period marked by intensive coal combustion in China. Since 2017, coal has been gradually replaced by natural gas for domestic heating during the cold season. Please change the title of the manuscript.

2. This manuscript only investigated the optical effects of organic aerosols. Please remove sentences regarding the health effects of organic aerosols from the Abstract and Introduction.

3. In this manuscript, coal combustion has been proposed as an important source. Please demonstrate the contribution of coal combustion areas without central heating during winter. Is coal combustion indeed a significant source?

4. Cl$^-$ is used as a marker of coal combustion. However, sea salt is also a significant source of Cl$^-$ in PM$_{2.5}$, particularly in cities like Shanghai and Guangzhou. Please ensure that only non-sea salt Cl$^-$ is included into the model. Please refer to the equation for calculating non-sea salt Cl$^-$ as provided in Ma et al. (10.5194/acp-18-5607-2018).

5. The authors proposed a light-absorbing carbonaceous continuum. However, it is important to note that there may be overlaps between the different carbon components. The operational definition of carbon components also varies. The compounds the authors refer to as 'char BC' most likely belong to 'brown carbon' or 'humic like substances', and are unlikely to bias optically based BC

measurements in large cities. Please address this point.

6. Please double check for grammar. There are lots of grammatic errors in this manuscript.

7. Please shorten the titles of each section in Results and Discussion.

Specific comments:

Section of Materials and Methods:

1. Please clarify the year in which the samples were collected.

2. Why does Figure 1 display the average aerosol optical depth at 550 nm instead of other wavelengths?

3. Please describe the instrumental method of ion chromatography.

4. The random errors and rotational ambiguity of the source apportionment model should be estimated using the bootstrap model and the displacement model. Please provide the evaluation results.

Section of Results and Discussion

5. Please clarify whether the value after average is standard deviation or interquartile range. For instance, in Line 232-232, "The concentrations of WIOC ranged from 1.45 to 12.95 $\mu gC/m^3$, with an average of $3.64 \pm 2.53$ $\mu gC/m^3$ among the 10 cities (Figure 2a)."

6. BB in Line 279 and MW in Line 320 should be defined.

7. $p$ should be in italic.

Technical corrections:

Line 20: Water-insoluble "organic" carbon

Line 45: WIOC "is" primarily originated

Line 157: the relative standard deviation of what?

Line 251: were higher than those of non-HULIS (Figure "2"c).

Line 426: It is important

---

## Author Comment (AC1)

We deeply appreciate Reviewers for their constructive recommendation and the helpful suggestion. We have updated the manuscript following these comments and addressed all points raised. These comments are very helpful for improving our manuscript. Specific responses to each of the comments are provided below (review's comments in **black**, our responses in blue, details of the changes made to the manuscript in *blue font*). And the modifications in the revised manuscript with marks are marked in yellow. We are pleased to provide the revised manuscript and hope both Reviewers are satisfied with our responses.

**Response to Reviewer #2**

**1) Reviewer's general comment:**

**In this study, different fractions of organic carbon in ambient fine particulate matter (PM2.5) in ten Chinese cities and their optical effects were investigated. Results indicated that the optical effects of extractable organic carbon are mainly contributed by relatively hydrophobic fractions (i.e., water-insoluble organic carbon and humic-like substances). Both empirical indices and the source apportionment model indicate that aromatic compounds from primary emissions tend to exhibit a stronger light-absorbing capacity. This study can provide significant information on the chemical compositions and sources of brown carbon (BrC) for further mitigating the climate effects of PM2.5. I recommend accepting this manuscript if the following comments could be addressed in the revised version.**

Response: We appreciate Reviewer#2 professional review for our article. We have revised the manuscript to address the comments. Our responses to the specific comments and changes made in the manuscript are given below

**2) Reviewer's comment:**

**1. This study cannot represent the entire China, because only samples from urban areas were analyzed. All these samples were collected during 2013-2014, a period marked by intensive coal combustion in China. Since 2017, coal has been gradually replaced by natural gas for domestic heating during the cold season. Please change**

**the title of the manuscript.**

Response: We appreciate the constructive feedback provided by the reviewer. It is acknowledged that our sampling sites are primarily located at typical urban areas, and therefore, they may not fully represent the entirety of China. In response, we have amended the title of the manuscript to reflect this focus: *"The Wate -insoluble organic carbon in PM2.5 of typical Chinese urban areas: light-absorbing properties, potential sources, radiative forcing effects, and the possible light-absorbing continuum"*.

Furthermore, we agree with the reviewer's observation regarding the significant policy shifts initiated by the Chinese government since 2017, particularly in promoting clean energy policies and prohibiting open biomass burning. It is important to note that our samples were collected during the period of 2013-2014, predating these clean energy initiatives. Consequently, our findings are reflective of a time when coal combustion and biomass burning exerted considerable influence on the sampled air quality. Additionally, during the sampling period, a substantial contribution from secondary aerosols to particulate pollution was observed in China. Given the significance of biomass burning, coal combustion, and Secondary Organic Aerosols (SOA) as sources of BrC with strong light-absorbing capabilities, WIOC emerges as a major contributor to the light absorption of BrC. The sample sets collected during this timeframe offer a unique opportunity to elucidate which sources predominantly contribute to Chinese WIOC and which exhibit the strongest light absorption capacity.

**3) Reviewer's comment:**

**2. This manuscript only investigated the optical effects of organic aerosols. Please remove sentences regarding the health effects of organic aerosols from the Abstract and Introduction.**

Response: Good suggestion! We agree with reviewer that this study mainly focus on the light-absorbing properties of WIOC. We have removed health effects of organic aerosols in the Abstract, and rewritten the second paragraph in the Introduction to emphasize the main point of this study.

*The manuscript is revised as follows:*

**Lines 66 to 81:** *"According to water solubility, OC can be classified into two main categories: water-soluble OC (WSOC) and water-insoluble OC (WIOC). While WSOC has been extensively studied over the past decades, with investigations focusing on its sources, light-absorbing properties, and atmospheric processes (Bosch et al., 2014; Dasari et al., 2019; Mo et al., 2021; Wang et al., 2020; Wozniak et al., 2014). WIOC, which makes up large fraction of OC (~up to 80%) and a substantial portion of light absorption by BrC, has received comparatively less attention. WIOC exhibits a significantly higher light-absorbing capacity compared to WSOC, attributed to the enrichment of strong light-absorbing BrC chromophores in WIOC. For instance, certain strong BrC chromophores like polycyclic aromatic hydrocarbons (PAHs) and their derivatives, as well as high-molecular-weight oligomers, are water-insoluble (Huang et al., 2020; Kalberer et al., 2006; Xie et al., 2017). Indeed, Zhang et al. (2013) reported that the light absorption by methanol-extracted OC in Los Angeles was approximately 3 and 21 times higher than that by WSOC. Moreover, field observations indicate that WIOC exhibits greater recalcitrance during long-range transport processes compared to WSOC, leading to a longer lifetime for WIOC (Fellman et al., 2015; Kirillova et al., 2014; Wozniak et al., 2012). Given that WIOC represents a relatively long-lived OC component with a higher light-absorbing capacity, a comprehensive understanding of its sources and light-absorbing properties is imperative."*

**4) Reviewer's comment:**

**3. In this manuscript, coal combustion has been proposed as an important source. Please demonstrate the contribution of coal combustion areas without central heating during winter. Is coal combustion indeed a significant source?**

Response: Thank you for your constructive comment. As suggested by the reviewer, we have conducted further analysis to investigate the spatiotemporal dynamics of WIOC sources. Despite coal combustion and biomass burning (BB) contributing comparably to the annual WIOC levels (31.1% vs. 31.0%), notable spatial and seasonal variations were evident. As illustrated in Figure R1, As showed in Figure

R1, during cold seasons, coal combustion contributed, on average, 17.4% of WIOC in areas without central heating, a proportion significantly lower than that attributed to BB (54.2%). Conversely, in areas with central heating, coal combustion exhibited a much higher contribution than BB (56.2% vs. 14.8%) during cold seasons. These observations underscore the prominence of coal combustion and BB as primary sources of WIOC in areas with and without central heating, respectively, particularly during cold seasons.

[Figure]

Figure R1. The spatial (Aeras with/without central heating) and seasonal (warm/cold seasons) variation of sources contribution of WIOC in China

***The manuscript is revised as follows:***

**Lines 391 to 411:** *"The primary sources of WIOC were combustion sources, with coal combustion and BB averagely account for 31.1% and 31.0% of the WIOC, respectively. Although the contribution of coal combustion to WIOC was comparable to that of BB, both exhibited distinct spatial and seasonal variations. Specifically, during winter, coal combustion emerged as the dominant source of WIOC, accounting*

*for 48.4% of the total, likely driven by increased coal usage in areas with central heating. Indeed, coal combustion constituted the primary source of WIOC in areas with central heating during cold seasons (56.2%). In contrast, in areas without central heating, the contribution of BB surpassed that of coal combustion significantly (54.2% vs. 17.3%). Therefore, coal combustion and BB were identified as the predominant sources of WIOC in areas with and without central heating, respectively, during cold seasons. Compared to primary emissions sources, the contributions of the sources related to aging processes and nitrogen-induced secondary formation were relatively lower, accounting for 18.2% and 5.2% of the WIOC, respectively. That may be due to these two secondary sources are more enriched in water-soluble components (HULIS-C + non-HULIS-C). Actually, although the uncertainties of sources contribution of HULIS-C and non-HULIS-C resolved by PMF model may be high, the aging processes and nitrogen-related secondary formation contributed 10.1% and 20.2% to HULIS-C, and 18.3% and 21.6% to non-HULIS-C, respectively. In addition, during the summer, when both temperature and solar radiation intensity rise, the contributions from aging processes and BB increased to 39.3% and 41.3%, respectively. In spring, a significant fraction of WIOC was associated with dust/soil, reaching up to 28.8%. Specially, the dust/soil contribution was much higher in the aeras with central hearing than those without central heating. This is consistent with the fact that sandstorms from the Gobi desert that borders China and Mongolia ride springtime winds to affect the air quality of Northern China (Filonchyk et al., 2024).''*

**5) Reviewer's comment:**

**4. Cl- is used as a marker of coal combustion. However, sea salt is also a significant source of Cl- in PM2.5, particularly in cities like Shanghai and Guangzhou. Please ensure that only non-sea salt Cl- is included into the model. Please refer to the equation for calculating non-sea salt Cl- as provided in Ma et al. (10.5194/acp-18-5607-2018).**

Response: We appreciate the reviewer for highlighting this aspect and recommending the referenced article. We acknowledge the contribution of sea salt

aerosols to Cl⁻, particularly in coastal cities like Guangzhou and Shanghai, as evidenced in our study. Before conducting the PMF model, we assessed the contribution of sea salt Cl⁻ to the total Cl using the equation: non-sea salt Cl⁻ ([nss-Cl⁻] = [Cl⁻] − 1.17 × [Na⁺]). The analysis revealed that the contribution of sea salt Cl- to the total Cl was generally below ~7% in Guangzhou and Shanghai. Furthermore, we performed comparisons between the PMF results obtained using only non-sea salt Cl⁻ and those using total Cl⁻. We found that the PMF factors and their corresponding contributions did not exhibit significant changes (within 6%). Therefore, we concluded that there was no need to further modify the model by using only non-sea salt Cl⁻ in our study.

**6) Reviewer's comment:**

**5. The authors proposed a light-absorbing carbonaceous continuum. However, it is important to note that there may be overlaps between the different carbon components. The operational definition of carbon components also varies. The compounds the authors refer to as 'char BC' most likely belong to 'brown carbon' or 'humic like substances', and are unlikely to bias optically based BC measurements in large cities. Please address this point.**

Response: Good point! Indeed, the distinction between various carbonaceous components is based on a conceptual and operational definition and does not correspond in reality to a clear boundary. In this work, the WIOC, HULIS-C, and non-HULIS-C are well-defined based on their polarity, while the definition BC, including the char- and soot-BC, is more related to the thermal and optical properties. These are two different operational definitions, so there may be overlap between these carbonaceous components. A part of char-BC may show a chemical and physical behavior similar to high-molecular-weight OC compounds (e.g., HULIS), which indeed overlap with BrC. We have added a discussion to the revised manuscript to deal with this point raised by the reviewer.

***The manuscript is revised as follows:***

**Lines 605 to 617:** *"It is important to acknowledge that carbonaceous aerosols encompass a wide array of diverse components, exhibiting a continuum of physical and chemical properties. The distinction between these carbonaceous components, as discussed above, is primarily based on conceptual and operational definitions, rather than clear boundaries in reality. In other words, the classification of carbonaceous components in aerosols is highly dependent on operational criteria. In this study, on the one hand, the WIOC, HULIS-C, and non-HULIS-C are well-defined based on their polarity. On the other hand, the definition of BC, which includes char- and soot-BC, is more closely associated with thermal and optical properties. These operational definitions may lead to overlaps between different carbonaceous components. For instance. BB and coal combustion emit large amounts of large molecular weight soluble compounds, such as HULIS (e.g., HULIS), which may char and produce false char EC signals in the TOT analysis (Yu et al., 2002). Additionally, certain portions of char-BC may exhibit chemical and physical behaviors akin to high-molecular-weight OC compounds, thereby overlapping with BrC. Therefore, there is no a clear boundary for the carbonaceous components."*

**7) Reviewer's comment:**

**6. Please double check for grammar. There are lots of grammatic errors in this manuscript.**

Response: We sorry for the grammatic errors. We have let a native speaker help polish the language.

**8) Reviewer's comment:**

**7. Please shorten the titles of each section in Results and Discussion.**

Response: Thanks for your suggestions. We have shortened the titles of each section in the Results and Discussion.

**9) Reviewer's comment:**

**Specific comments:**

**Section of Materials and Methods:**

**Please clarify the year in which the samples were collected.**

Response: Thanks for your comment. We have clarified that all the filter samples were collected during 2013 to 2014 in the revised manuscript.

*The manuscript is revised as follows:*

**Lines 104 to 106:** *"All the filter samples were collected on pre-combusted (450°C, 6h) quartz-fiber filter (Pall, England) from 2013 to 2014, use a high-volume sampler at a flow rate of ~ 1000L/min."*

**10) Reviewer's comment:**

**Why does Figure 1 display the average aerosol optical depth at 550 nm instead of other wavelengths?**

Response: Thanks for your questions. Aerosol optical depth (AOD), is indeed often measured at a wavelength of 550 nanometers (nm). This wavelength is chosen because it falls within the visible spectrum of light and is sensitive to changes in aerosol concentration in the atmosphere. Regarding the sensitivity, Aerosols, such as dust, smoke, and pollutants, have a significant impact on the scattering and absorption of sunlight. At 550 nm, the scattering and absorption properties of common aerosols are well-characterized, making it an ideal wavelength for measuring their optical thickness. In addition, many satellite sensors designed for measuring AOD, such as the Moderate Resolution Imaging Spectroradiometer (MODIS), operate at or around 550 nm. This wavelength was chosen for its practicality in remote sensing applications. Finally, using a standard wavelength like 550 nm allows for comparability between different studies and datasets. It provides a common reference point for researchers to assess aerosol loading across different regions and times. Therefore, 550 nm is the most common wavelength used for AOD measurements.

**11) Reviewer's comment:**

**Please describe the instrumental method of ion chromatography.**

Response: We have provided the instrumental details of ion chromatography in the Supporting information of the revised manuscript (Text S1). Briefly, 3 anions ($Cl^-$, $NO_3^-$ and $SO_4^{2-}$) and 4 cations ($Na^+$, $K^+$, $Ca^{2+}$ and $NH_4^+$) were analyzed with ion-chromatography (761 Compact IC, Metrohm, Switzerland). Anions were separated on a Metrohm Metrosep A sup5-250 column with 3.2 mM $Na_2CO_3$ and 1.0 mM $NaHCO_3$ as the eluent and 35 mM $H_2SO_4$ for a suppressor. Cations were measured using a Metrohm Metrosep C4-150 column with 2 mM sulfuric acid as the eluent. The injection loop volume for anion and cation was 100 μL. The water-soluble ions analyses were duplicated for several filter samples, and the overall relative standard deviations were generally less than 4%.

**12) Reviewer's comment:**

**The random errors and rotational ambiguity of the source apportionment model should be estimated using the bootstrap model and the displacement model. Please provide the evaluation results.**

Response: Thanks for your insightful suggestions. We have evaluated the estimate the errors associated with both random and rotational ambiguity of PMF solution by the bootstrap (BS) model and the displacement (DISP) model. The BS factors from the resampled data matrices are mapped to the base run factors to provide the reproducibility of different base run factors due to the random errors. The 4 BS factors model showed that factor mapping higher than 85%, indicating both uncertainties and the number of factors were appropriate. DISP mainly explores rotational ambiguity in the PMF results. At 4 factor PMF model, no swaps were found in DISP model. In the revised manuscript, we have provided the uncertainty assessment of the PMF model.

***The manuscript is revised as follows:***

**Lines 205 to 212:** *"The errors associated with both random and rotational ambiguity in the PMF solution were assessed using the bootstrap (BS) model and the displacement (DISP) model. The BS model involves estimating errors by resampling data matrices, with the resulting BS factors being aligned with the base run factors to*

*gauge the reproducibility of different factors amidst random errors. Analysis using a 4-factor BS model indicated a factor mapping exceeding 85%, suggesting both the suitability of the number of factors and the presence of uncertainties. On the other hand, DISP primarily investigates rotational ambiguity within the PMF outcomes. Notably, in the context of a 4-factor PMF model, no swaps were identified in the DISP analysis."*

**2) Reviewer's comment:**

**Section of Results and Discussion**

**Please clarify whether the value after average is standard deviation or interquartile range. For instance, in Line 232-232, "The concentrations of WIOC ranged from 1.45 to 12.95 µgC/m3, with an average of 3.64 ± 2.53 µgC/m3 among the 10 cities (Figure 2a)."**

Response: Thanks for pointing this out. The value after average is the one standard deviation. We have clarified this point in the revised manuscript.

**2) Reviewer's comment:**

**BB in Line 279 and MW in Line 320 should be defined.**

Response: We thank the reviewers for their meticulous work. We have defined the BB (biomass burning) in the right position the revised manuscript. The MW (molecular weight) was defined before Line 320 in the initial manuscript, so we do not define it again in the Line 320.

**2) Reviewer's comment:**

**p should be in italic.**

Response: Corrected.

**2) Reviewer's comment:**

**Technical corrections:**

**Line 20: Water-insoluble "organic" carbon**

Response: Corrected.

**2) Reviewer's comment:**

**Line 45: WIOC "is" primarily originated**

    Response: Corrected.

**2) Reviewer's comment:**

**Line 157: the relative standard deviation of what?**

    Response: It should be *"The carbon contents of WIOC were determined by an OC/EC analyzer with standard deviation of reproducibility test less than 3%"* (**Lines 151 to 152**)

*The manuscript is revised as follows:*

**2) Reviewer's comment:**

**Line 251: were higher than those of non-HULIS (Figure "2"c).**

    Response: Corrected

*The manuscript is revised as follows:*

**2) Reviewer's comment:**

**Line 426: It is important**

    Response: Corrected

---

## Author Comment (AC2)

We deeply appreciate Reviewers for their constructive recommendation and the helpful suggestion. We have updated the manuscript following these comments and addressed all points raised. These comments are very helpful for improving our manuscript. Specific responses to each of the comments are provided below (review's comments in **black**, our responses in blue, details of the changes made to the manuscript in *blue font*). And the modifications in the revised manuscript with marks are marked in yellow. We are pleased to provide the revised manuscript and hope both Reviewers are satisfied with our responses.

**Response to Reviewer #1**

**1) Reviewer's general comment:**

**The manuscript provided a detailed study on the light-absorbing characteristics, potential sources, and radiative forcing effects of WIOC across ten cities in China. The authors compared the characteristics of extractable OC (EX-OC) with different polarities (WIOC, HULIS, and non-HULIS). They discovered that biomass burning and coal combustion are the primary sources of WIOC in China and radiative forcing of EX-OC is mainly due to WIOC and HULIS. Furthermore, this study proposed a hypothesis of a light-absorbing carbonaceous continuum, demonstrating that carbonaceous components from fossil sources tend to have a stronger light-absorbing capacity, higher aromatic levels, and greater recalcitrance in the atmosphere. Overall, the paper is well-structured and well-written, but there are still some areas that require improvement. Please see my comments below.**

Response: We thank the Reviewer#1 for carefully reviewing our manuscript and for the insightful comments and the constructive feedback! Please find our answers to each individual point below:

**2) Reviewer's comment:**

**Major Comments:**

**In this study, the authors divide China into areas with and without central heating, which is a meaningful approach. However, the division based on a simple line in Figure 1 lacks convincing evidence. Are there data to support this division?**

**Moreover, the sources and light absorption properties of WIOC are expected to differ significantly between areas with and without central heating. However, the article does not adequately discuss the spatial differences in WIOC sources and light absorption capabilities. The authors should address this aspect.**

Response: Thank you for your helpful suggestions. The areas with central heating typically experience an average annual temperature below 15°C, while areas without central heating generally have higher average annual temperatures, often exceeding 15°C. Notably, the differentiation between areas with and without central heating in our study closely aligns with the well-recognized Qinling Mountains-Huaihe River line, which serves as the north-south boundary in China (Figure R1). We have incorporated additional details and descriptions for Figure 1 to provide further clarity.

[Figure]

**Figure R1.** The central heating north–south boundary in China.

We have incorporated additional discussions on the spatial differences in the light-absorbing capacity and sources of WIOC in the revised manuscript, as reviewer suggested. In terms of light-absorbing capacity, our findings reveal a significant

disparity in the $MAE_{365}$ of WIOC between areas with and without central heating. Specifically, areas with central heating exhibited a notably higher $MAE_{365}$ (1.75 ± 0.64 $m^2$/gC) compared to those without (1.48 ± 0.46 $m^2$/gC), with a statistically significant difference (p < 0.01). This disparity was more pronounced during colder seasons, with a 21.5% difference in $MAE_{365}$ between areas with central heating (2.20 ± 0.51 $m^2$/gC) and those without (1.81 ± 0.28 $m^2$/gC), compared to a 10.3% difference during warmer seasons. We attribute this spatial variability primarily to coal combustion, given the considerably higher coal consumption for central/domestic heating in areas with central heating.

Regarding the sources of WIOC, while the contribution of coal combustion was comparable to that of BB, both exhibited distinct spatial and seasonal variations. During winter, coal combustion emerged as the dominant source, constituting 48.4% of the total WIOC, likely due to increased coal usage in areas with central heating. Notably, coal combustion accounted for 56.2% of WIOC in areas with central heating during cold seasons, whereas in areas without central heating, BB surpassed coal combustion significantly (54.2% vs. 17.3%). Thus, coal combustion and BB were identified as the predominant sources of WIOC in areas with and without central heating, respectively, during cold seasons.

Moreover, during summer, with rising temperatures and solar radiation intensity, contributions from aging processes and BB increased to 39.3% and 41.3%, respectively. In spring, a significant fraction of WIOC was associated with dust/soil, reaching up to 28.8%, particularly pronounced in areas with central heating. This observation aligns with the influence of sandstorms from the Gobi desert, which border China and Mongolia, affecting the air quality of Northern China during springtime (Filonchyk et al., 2024).

***The manuscript is revised as follows:***

**Lines 301 to 308 :** *"Spatially, the MAE365 of WIOC was significantly higher in areas with central heating than without central heating (1.75 ± 0.64 m2/gC vs. 1.48 ± 0.46 m2/gC, p < 0.01). The difference in MAE365 of WIOC between the areas with and*

*without central heating was more pronounced during colder seasons (2.20 ± 0.51 m2/gC vs. 1.81 ± 0.28 m2/gC, 21.5% difference) than warmer seasons (1.29 ± 0.37 m2/gC vs. 1.17 ± 0.26 m2/gC, 10.3% difference). Given that coal consumption for central/domestic heating is considerably higher in areas with central heating compared to those without, it is plausible that the spatial variability in MAE365 of WIOC is predominantly influenced by coal combustion."*

**Lines 390 to 411:** *"Although the contribution of coal combustion to WIOC was comparable to that of BB, both exhibited distinct spatial and seasonal variations (Figure 4a). Specifically, during winter, coal combustion emerged as the dominant source of WIOC, accounting for 48.4% of the total, likely driven by increased coal usage in areas with central heating. Indeed, coal combustion constituted the primary source of WIOC in areas with central heating during cold seasons (56.2%). In contrast, in areas without central heating, the contribution of BB surpassed that of coal combustion significantly (54.2% vs. 17.3%). Therefore, coal combustion and BB were identified as the predominant sources of WIOC in areas with and without central heating, respectively, during cold seasons. Compared to primary emissions sources, the contributions of the sources related to aging processes and nitrogen-induced secondary formation were relatively lower, accounting for 18.2% and 5.2% of the WIOC, respectively. That may be due to these two secondary sources are more enriched in water-soluble components (HULIS-C + non-HULIS-C). Actually, although the uncertainties of sources contribution of HULIS-C and non-HULIS-C resolved by PMF model may be high, the aging processes and nitrogen-related secondary formation contributed 10.1% and 20.2% to HULIS-C, and 18.3% and 21.6% to non-HULIS-C, respectively. In addition, during the summer, when both temperature and solar radiation intensity rise, the contributions from aging processes and BB increased to 39.3% and 41.3%, respectively. In spring, a significant fraction of WIOC was associated with dust/soil, reaching up to 28.8%. Specially, the dust/soil contribution was much higher in the aeras with central hearing than those without central heating. This is consistent with the fact that sandstorms from the Gobi desert that borders China and Mongolia ride springtime winds to affect the air quality of Northern China*

*(Filonchyk et al., 2024)."*

**3) Reviewer's comment:**

**The method for extracting WIOC involves methanol extraction, concentrated, and analysis using an OC/EC analyzer, while WSOC is determined using a liquid TOC analyzer. The analysis mechanism of OC/EC analyzers and TOC analyzers differ, including the different catalysts and detectors they employ. Comparing measurement results from these two methods may be challenging. Additionally, the authors did not provide the blank value for OC/EC determination of WIOC, which is crucial given methanol's high propensity for extracting atmospheric organic matter.**

Response: Thanks for the good questions! As highlighted by the reviewer, it's important to note that the analytical procedure and mechanism for determining carbon content differ between the OC/EC analyzer and the TOC analyzer. In our study, the WIOC is extracted and dissolved in methanol, making direct determination of its carbon content by the TOC analyzer unfeasible. Consequently, comparisons between the determined WIOC content using these two methods are not possible. However, it is important to note that both instruments involve converting carbon in the sample to $CO_2$ and detecting $CO_2$ using a nondispersive infrared detector (NDIR) or a flame ionization detector (FID) after $CO_2$ conversion to methane. Thus, the analytical mechanisms are similar. Indeed, previous studies have systematically compared these two methods for determining WSOC in aerosols. The results have demonstrated no significant differences between the measurements obtained from the two methods(Fan et al., 2012). Therefore, the carbon content of various carbonaceous components determined by these two methods should be comparable. We have elaborated further on this point in the revised manuscript.

The field blanks for the WSOC, HULIS-C and WIOC were $0.39 \pm 0.16$, $0.66 \pm 0.21$, and $1.75 \pm 0.48$ ug $C/cm^2$, respectively. All WSOC, HULIS, and WIOC concentrations presented in this study were corrected with field blanks.

*The manuscript is revised as follows:*

**Lines 152 to 162:** *"The analysis mechanism of OC/EC analyzers and TOC analyzers differ, including the different catalysts and detectors they employ. However, it is important to note that both OC involve converting carbon in the sample to $CO_2$ and detecting $CO_2$ using a nondispersive infrared detector (NDIR) or a flame ionization detector (FID) after $CO_2$ conversion to methane. Thus, the analytical mechanisms are similar. Indeed, previous studies have systematically compared these two methods for determining WSOC in aerosols (Yu et al., 2002). The results have demonstrated no significant differences between the measurements obtained from the two methods. Therefore, the carbon content of various carbonaceous components determined by these two methods should be comparable. Based on extractable OC (EX-OC) polarity, the EX-OC was separated into WIOC, HULIS-C, and non-HULIS-C. All WSOC, HULIS-C, and WIOC concentrations presented in this study were corrected with field blanks ($0.39 \pm 0.16$, $0.66 \pm 0.21$, and $1.75 \pm 0.48$ $\mu gC/cm^2$, respectively)."*

**4) Reviewer's comment:**

**The authors primarily utilize the PMF model to quantify the sources of WIOC. The authors suggest that the increase in coal combustion led to an increase in Abs365 in overall EX-OC (line 407-411), which seems plausible. However, correlating the quantitative results from coal combustion predicted by PMF with the Abs365 of overall EX-OC (Figure S3c) is not appropriate. The authors quantified the sources of WSOC using dual carbon isotopes ($\delta13C$ and $\Delta14C$) in their previous study. Comparing the PMF model with source apportionment results based on dual carbon isotopes presents challenges. While 14C can accurately distinguish fossil sources from non-fossil sources, without chemical markers, it's difficult to quantitatively analyze sources related to atmospheric processes (such as secondary sources). On the other hand, although PMF can theoretically incorporate various chemical markers and include secondary sources in quantitative analysis, its results involve more human interpretation factors and**

**may not be as precise as 14C. Thus, caution is warranted when comparing source apportionment results obtained from PMF and dual carbon isotopes.**

Response: Thank you for raising this important point. We fully concur that attempting to establish a correlation between the quantitative results of WIOC from coal combustion predicted by PMF and the $Abs_{365}$ of overall EX-OC is inappropriate. In our previous study, we utilized dual carbon isotopes to quantify the source of WSOC, while the sources of WIOC were determined using the PMF model. As noted by the reviewer, 14C isotopic analysis offers a precise means of distinguishing between fossil and non-fossil sources. Conversely, while the PMF model theoretically has the capacity to resolve a broader range of source types than 14C isotopic analysis, its results are subject to more human interpretation factors and may not be as definitive as those obtained through 14C analysis. In light of this insight, we have removed the discussion on the relationship between $Abs_{365}$ of EX-OC and coal combustion contribution predicted by the PMF model in the revised manuscript.

**5) Reviewer's comment:**

**In Section 3.4, the authors propose a concept of a light-absorbing carbonaceous aerosol continuum, which I find intriguing. It's important to note that in this study, WIOC, HULIS, and non-HULIS are well-defined based on their polarity. These components indeed exhibit significant differences in both sources and light absorption properties. However, regarding BC, I contend that BC is the strongest light-absorbing component. Although this assertion isn't reflected in Figure 6, the authors further subdivide BC into char and soot in this section without providing a clear definition. In reality, char and soot are defined differently across various environmental matrices (https://doi.org/10.1038/s43017-022-00316-6). For aerosols, biomass burning and coal combustion emit large amounts of large molecular weight soluble compounds, which may char and produce false char EC signals (artifacts). Additionally, there are overlaps between char and BrC. These issues warrant attention and clarification from the authors.**

Response: We appreciate the insightful comments provided by the reviewer. It is indeed acknowledged that the definitions of Char and Soot vary across different environmental matrices (Coppola et al., 2022). In our study, the distinction between char-BC and soot-BC within carbonaceous aerosols is primarily based on operational definitions derived from the Interagency Monitoring of Protected Visual Environments (IMPROVE) protocol and the thermal-optical reflectance (TOR) method for OC and EC analysis, as referenced in previous studies (Cai et al., 2023; Han et al., 2010).

The carbonaceous aerosols encompass a wide array of diverse components, exhibiting a continuum of physical and chemical properties. The differentiation between these components relies on conceptual and operational definitions, which may not necessarily correspond to clear boundaries in reality. In our work, WIOC, HULIS-C, and non-HULIS-C are defined based on their polarity, while the definition of BC, including char-BC and soot-BC, is primarily related to thermal and optical properties. However, it is important to acknowledge that these operational definitions may result in overlaps between different carbonaceous components. For instance, carbonaceous emissions from BB and coal combustion often contain large molecular weight soluble compounds, such as HULIS, which can char and produce false char EC signals in the TOR analysis (Yu et al., 2002). Additionally, certain fractions of char-BC may exhibit chemical and physical behaviors resembling those of high-molecular-weight OC compounds, potentially overlapping with BrC. Therefore, delineating clear boundaries for carbonaceous components remains challenging.

***The manuscript is revised as follows:***

**Lines 497 to 502:** *"Char and soot are defined differently across various environmental matrices (Coppola et al., 2022). For the carbonaceous aerosols, the char-BC and soot-BC are widely and operationally defined by different temperatures in the Interagency Monitoring of Protected Visual Environments (IMPROVE) protocol and the thermal-optical reflectance (TOR) method for OC and EC analysis (Cai et al., 2023; Han et al., 2010). Generally, the light-absorbing capacity of soot-BC is higher*

*than char-BC (Andreae and Gelencser, 2006; Corbin et al., 2019; Schnaiter et al., 2003)."*

**Lines 605 to 617:** *"It is important to acknowledge that carbonaceous aerosols encompass a wide array of diverse components, exhibiting a continuum of physical and chemical properties. The distinction between these carbonaceous components, as discussed above, is primarily based on conceptual and operational definitions, rather than clear boundaries in reality. In other words, the classification of carbonaceous components in aerosols is highly dependent on operational criteria. In this study, on the one hand, the WIOC, HULIS-C, and non-HULIS-C are well-defined based on their polarity. On the other hand, the definition of BC, which includes char- and soot-BC, is more closely associated with thermal and optical properties. These operational definitions may lead to overlaps between different carbonaceous components. For instance. BB and coal combustion emit large amounts of large molecular weight soluble compounds, such as HULIS (e.g., HULIS), which may char and produce false char EC signals in the TOT analysis (Yu et al., 2002). Additionally, certain portions of char-BC may exhibit chemical and physical behaviors akin to high-molecular-weight OC compounds, thereby overlapping with BrC. Therefore, there is no a clear boundary for the carbonaceous components."*

**6) Reviewer's comment:**
**Minor Comments:**
**Line 20: "water-insoluble carbon" change to "water-insoluble organic carbon"**
    Response: Corrected

**7) Reviewer's comment:**
**Line 145 -147, the same phrase appears twice.**
    Response: Thanks for your careful work. We have deleted the identical sentences.

**8) Reviewer's comment:**
**Line 148, the authors mention using SPE for HULIS extraction. Given the various methods available for HULIS isolation (as referenced in https://doi.org/10.1016/j.envpol.2013.05.055), it would be helpful to provide reasons for choosing SPE for HULIS isolation.**

Response: Thanks for your comment. The HULIS is an operational definition, and there are many methods to isolate the HULIS. The reason we select the HLB-SPE column for HULIS isolation is because the HLB-SPE column showed excellent reproducibility and high recovery yield (Fan et al., 2012; Lin et al., 2010).

*The manuscript is revised as follows:*

**Lines 139 to 147:** *"HULIS are operationally defined by the procedure used for isolation from bulk WSOC by removing low molecular weight organic acids and inorganic ions. The HLB (Oasis, 30 µm, 60 mg/cartridge, Waters, USA)-SPE method is most widely used to isolate HULIS due to its excellent reproducibility and high recovery yield (Fan et al., 2012; Lin et al., 2010b). Therefore, we used an HLB-SPE column to isolate the HULIS"*

**9) Reviewer's comment:**

**Line 271-272, "BrC" changed to "extractable OC". There is no accurate method for extracting BrC.**

Response: Corrected.

**10) Reviewer's comment:**

**Line 388, it should be "carbon mass contribution"**

Response: Corrected.

**11) Reviewer's comment:**

**Figure 6. I suggest emphasizing in the caption that WIOC tends to be OC soluble in methanol but not in water. The WIOC in this work is not real water-insoluble organic carbon.**

Response: Good suggestion! We have emphasized that *"The WIOC in this study does not strictly denote OC that is insoluble in water, it is more likely to be OC that is insoluble in water but soluble in methanol."* in the Figure 6.

**References:**

Andreae, M.O., Gelencser, A. (2006), Black carbon or brown carbon? The nature of light-absorbing carbonaceous aerosols, *Atmospheric Chemistry and Physics*, *6*, 3131-3148.https://doi.org/10.5194/acp-6-3131-2006

Cai, J., Jiang, H., Chen, Y., Liu, Z., Han, Y., Shen, H., Song, J., Li, J., Zhang, Y., Wang, R., Chen, J., Zhang, G. (2023), Char dominates black carbon aerosol emission and its historic reduction in China, *Nature Communications*, *14*(1), 6444.https://doi.org/10.1038/s41467-023-42192-8

Coppola, A.I., Wagner, S., Lennartz, S.T., Seidel, M., Ward, N.D., Dittmar, T., Santín, C., Jones, M.W. (2022), The black carbon cycle and its role in the Earth system, *Nature Reviews Earth & Environment*, *3*(8), 516-532.https://doi.org/10.1038/s43017-022-00316-6

Corbin, J.C., Czech, H., Massabò, D., de Mongeot, F.B., Jakobi, G., Liu, F., Lobo, P., Mennucci, C., Mensah, A.A., Orasche, J., Pieber, S.M., Prévôt, A.S.H., Stengel, B., Tay, L.L., Zanatta, M., Zimmermann, R., El Haddad, I., Gysel, M. (2019), Infrared-absorbing carbonaceous tar can dominate light absorption by marine-engine exhaust, *Npj Climate and Atmospheric Science*, *2*.https://doi.org/10.1038/s41612-019-0069-5

Fan, X., Song, J., Peng, P.a. (2012), Comparison of isolation and quantification methods to measure humic-like substances (HULIS) in atmospheric particles, *Atmospheric Environment*, *60*, 366-374.https://doi.org/10.1016/j.atmosenv.2012.06.063

Filonchyk, M., Peterson, M.P., Zhang, L., Yan, H. (2024), An analysis of air pollution associated with the 2023 sand and dust storms over China: Aerosol properties and PM10 variability, *Geoscience Frontiers*, *15*(2), 101762.https://doi.org/https://doi.org/10.1016/j.gsf.2023.101762

Han, Y.M., Cao, J.J., Lee, S.C., Ho, K.F., An, Z.S. (2010), Different characteristics of char and soot in the atmosphere and their ratio as an indicator for source identification in Xi'an, China, *Atmos. Chem. Phys.*, *10*(2), 595-607.https://doi.org/10.5194/acp-10-595-2010

Lin, P., Engling, G., Yu, J.Z. (2010), Humic-like substances in fresh emissions of rice straw burning and in ambient aerosols in the Pearl River Delta Region, China, *Atmospheric Chemistry and Physics*, *10*(14), 6487-6500.https://doi.org/10.5194/acp-10-6487-2010

Schnaiter, M., Horvath, H., Möhler, O., Naumann, K.H., Saathoff, H., Schöck, O.W. (2003), UV-VIS-NIR spectral optical properties of soot and soot-containing aerosols, *Journal of Aerosol Science*, *34*(10), 1421-1444.https://doi.org/10.1016/s0021-8502(03)00361-6

Yu, J.Z., Xu, J., Yang, H. (2002), Charring Characteristics of Atmospheric Organic Particulate Matter in Thermal Analysis, *Environmental Science & Technology*, *36*(4), 754-761.https://doi.org/10.1021/es015540q

---

## Author Comment (AC3)

We deeply appreciate Reviewers for their constructive recommendation and the helpful suggestion. We have updated the manuscript following these comments and addressed all points raised. These comments are very helpful for improving our manuscript. Specific responses to each of the comments are provided below (review's comments in **black**, our responses in blue, details of the changes made to the manuscript in *blue font*). And the modifications in the revised manuscript with marks are marked in yellow. We are pleased to provide the revised manuscript and hope both Reviewers are satisfied with our responses.

**Response to Reviewer #3**

**1) Reviewer's general comment:**

**Mo et al. investigated light absorption and the sources of extractable organic carbon (EX-OC), which encompasses water-insoluble carbon (WIOC), humic-like substances (HULIS-C), and hydrophilic OC (non-HULIS-C). The study revealed that WIOC constituted the majority of OC mass concentrations and light-absorbing efficiency at 365 nm. Additionally, the authors found that the radiative forcing effects of EX-OC were mainly contributed by relatively hydrophobic fractions. The authors also proposed a light-absorbing carbonaceous continuum, revealing that components enriched with fossil sources exhibit stronger light-absorbing capacity, higher aromatic levels, increased molecular weights, and greater atmospheric recalcitrance. This study is pivotal for comprehensively understanding the climate-forcing brown carbon and developing related mitigating strategies. However, before publication, the authors need to address the following concerns:**

Response: We thank Reviewer#3 for the assessment of our manuscript and for providing suggestions to improve clarity and quality.

**2) Reviewer's comment:**

**Line 25–27: Too many brackets in this sentence. Please simplify this sentence to create a concise abstract.**

Response: Thanks for your suggestion. This sentence has been revised to *"On average, WIOC made up 33.4 ± 7.66% and 40.5 ± 9.73% of concentrations and light*

*absorption at 365 nm (Abs365) of extractable OC (EX-OC), which includes relatively hydrophobic OC (WIOC and humic-like substances: HULIS-C) and hydrophilic OC (non-humic-like substances: non-HULIS-C)."* (**Lines 25 to 28**)

**3) Reviewer's comment:**

**Line 51: Please review and remove any duplicate abbreviation definitions found throughout the manuscript, such as OC, WIOC, etc.**

Response: We have removed the duplicate abbreviation definitions in the revised manuscript.

**4) Reviewer's comment:**

**Lines 65-91: This paragraph lacks clarity in addressing the main questions concerning WIOC. While the author discusses the method of measuring light-absorbing OC and compares the light-absorbing properties of WIOC and WSOC, the discussion on the health effects and atmospheric lifetime of WIOC seems disconnected. Additionally, the motivation for this study is relatively vague. Therefore, I would suggest that the authors restructure the paragraph to clearly emphasize current scientific inquiries and provide a more cohesive rationale for their research.**

Response: We appreciate the reviewer's insightful comments. In response, we have reconstructed this paragraph to emphasize current scientific inquiries and provide a more cohesive rationale for our research. Please see the revised manuscript

***The manuscript is revised as follows:***

**Lines 66 to 81:** *"According to water solubility, OC can be classified into two main categories: water-soluble OC (WSOC) and water-insoluble OC (WIOC). While WSOC has been extensively studied over the past decades, with investigations focusing on its sources, light-absorbing properties, and atmospheric processes (Bosch et al., 2014; Dasari et al., 2019; Mo et al., 2021; Wang et al., 2020; Wozniak et al., 2014). WIOC, which makes up large fraction of OC (~up to 80%) and a substantial portion of light*

*absorption by BrC, has received comparatively less attention. WIOC exhibits a significantly higher light-absorbing capacity compared to WSOC, attributed to the enrichment of strong light-absorbing BrC chromophores in WIOC. For instance, certain strong BrC chromophores like polycyclic aromatic hydrocarbons (PAHs) and their derivatives, as well as high-molecular-weight oligomers, are water-insoluble (Huang et al., 2020; Kalberer et al., 2006; Xie et al., 2017). Indeed, Zhang et al. (2013) reported that the light absorption by methanol-extracted OC in Los Angeles was approximately 3 and 21 times higher than that by WSOC. Moreover, field observations indicate that WIOC exhibits greater recalcitrance during long-range transport processes compared to WSOC, leading to a longer lifetime for WIOC (Fellman et al., 2015; Kirillova et al., 2014; Wozniak et al., 2012). Given that WIOC represents a relatively long-lived OC component with a higher light-absorbing capacity, a comprehensive understanding of its sources and light-absorbing properties is imperative."*

**5) Reviewer's comment:**

**Lines 145-147: Please review the two identical sentences and revise them accordingly.**

Response: Thank you for your careful review. We have addressed the repeated sentences in the revised manuscript

**6) Reviewer's comment:**

**Line 251: Figure 1c should be Figure 2c?**

Response: Corrected.

**7) Reviewer's comment:**

**Lines 262-304: A more comprehensive discussion would involve comparing the light-absorbing properties of EX-OC between areas with and without central heating.**

Response: Thanks for your good suggestion. We have added more discussion on

the spatial variation in light-absorbing properties of EX-OC in the revised manuscript. We found that a significant disparity in the $MAE_{365}$ of WIOC between areas with and without central heating. Specifically, areas with central heating exhibited a notably higher $MAE_{365}$ ($1.75 \pm 0.64$ m$^2$/gC) compared to those without ($1.48 \pm 0.46$ m$^2$/gC), with a statistically significant difference ($p < 0.01$). This disparity was more pronounced during colder seasons, with a 21.5% difference in MAE365 between areas with central heating ($2.20 \pm 0.51$ m$^2$/gC) and those without ($1.81 \pm 0.28$ m$^2$/gC), compared to a 10.3% difference during warmer seasons. We attribute this spatial variability primarily to coal combustion, given the considerably higher coal consumption for central/domestic heating in areas with central heating.

*The manuscript is revised as follows:*

**Lines 301 to 308:** *"Spatially, the MAE365 of WIOC was significantly higher in areas with central heating than without central heating ($1.75 \pm 0.64$ m2/gC vs. $1.48 \pm 0.46$ m2/gC, p < 0.01). The difference in MAE365 of WIOC between the areas with and without central heating was more pronounced during colder seasons ($2.20 \pm 0.51$ m2/gC vs. $1.81 \pm 0.28$ m2/gC, 21.5% difference) than warmer seasons ($1.29 \pm 0.37$ m2/gC vs. $1.17 \pm 0.26$ m2/gC, 10.3% difference). Given that coal consumption for central/domestic heating is considerably higher in areas with central heating compared to those without, it is plausible that the spatial variability in MAE365 of WIOC is predominantly influenced by coal combustion."*

**8) Reviewer's comment:**

**Lines 269-272: Given that the light absorption of BrC, as measured by solvent extraction, appears to be underestimated compared to under ambient aerosol conditions, it is imperative to determine whether the authors considered this factor when comparing with "tar ball" and "unextractable dark BrC".**

Response: Indeed, the light absorption of BrC measured in solvent extracts often underestimates its absorption in ambient aerosol conditions. To address this disparity and predict the corresponding BrC absorption in ambient aerosols, it's essential to

calibrate the absorption determined in solvent extracts using a correction factor. To date, the correction factor proposed by Liu et al. (2013), commonly set at 2, is widely used for this purpose. Even after applying this correction factor, we observed that the MAE550 of WIOC ($0.28 \pm 0.09$ m$^2$/gC) remains an order of magnitude lower than that of "tar ball" ($\sim$3.6 to 4.1 m$^2$/g) and "unextractable dark BrC" ($\sim$1.2 m$^2$/g). In response, we have enriched the discussion on BrC light absorption correction in the revised manuscript.

***The manuscript is revised as follows:***

**Lines 279 to 289:** *"It should be noted that light absorption of BrC, as measured by solvent extraction, appears to be underestimated compared to under ambient aerosol conditions. To accurately derive the corresponding BrC absorption in ambient aerosols, it is necessary to calibrate the absorption determined in solvent extracts using a correction factor. Presently, the correction factor proposed by Liu et al. (2013), typically set at 2, is widely employed for this purpose. Despite WIOC being recognized as the most light-absorbing OC component, even after applying this correction factor, we observed that the MAE of WIOC at 550 nm($0.28 \pm 0.09$ m$^2$/gC) remains an order of magnitude lower than that of amorphous tar ball BrC (approximately 3.6 to 4.1 m$^2$/g) and unextractable "dark BrC" (approximately 1.2 m$^2$/g) as determined by transmission electron microscopy (Alexander et al., 2008; Chakrabarty et al., 2023), indicating the light-absorbing capacity of the extractable OC is relatively weakly."*

**9) Reviewer's comment:**

**Line 327: Please maintain consistency to ensure a unified description of E2/E3 in both the plot and text.**

Response: Corrected

**10) Reviewer's comment:**

**Lines 333-335 and 341-342: The authors observed a robust correlation between WIOC and Abs365, WIOC (r = 0.97, p < 0.01), indicating similar sources and**

**formation processes. However, they also concluded differences in sources and formation processes between WIOC and light-absorbing compounds during warm seasons. These findings may appear contradictory and confusing. Clarification is needed regarding the rationale behind analyzing the correlations between WIOC and Abs365, WIOC in individual warm and cold seasons.**

Response: Thank you for your constructive comments. We apologize for the lack of clarity in our previous sentences. Our intention was to show that while WIOC correlates well with Abs365, WIOC throughout the entire year ($r = 0.97$, $p < 0.01$), as indicated in Table S2, the correlations of WIOC with water-soluble ions vary notably between warm and cold seasons. This discrepancy suggests differences in sources and formation processes between WIOC and light-absorbing compounds during warm and cold seasons. We have revised these sentences to enhance clarity in the revised manuscript.

*The manuscript is revised as follows:*

**Lines 356 to 360:** *"Although the WIOC exhibited strong correlation with $Abas_{365, WIOC}$ ($r = 0.97$, $p < 0.01$) for the entire year, as listed in Table S2, the correlations between WIOC and Abs365, WIOC, as well as WIOC and water-soluble ions, differed notably between warm and cold seasons. This discrepancy suggests differences in sources and formation processes of WIOC and light-absorbing compounds between warm and cold season."*

**11) Reviewer's comment:**

**Line 375: Figure 3b should be Figure 4b?**

Response: Corrected.

**12) Reviewer's comment:**

**Line 382: Please clarify the methodology employed by the authors to determine the source contributions to Abs365, WIOC.**

Response: Thanks for your suggestion. We used the PMF model to resolve the sources contribution to $Abs_{365, WIOC}$. This methodology has been clarified in the revised manuscript.

*The manuscript is revised as follows:*

**Lines 413 :** *"Figure 4c shows the contributions of sources identified by PMF model to the $Abs_{365, WIOC}$."*

**13) Reviewer's comment:**

**Lines 390-392: Could you elaborate on why biomass burning, rather than secondary sources, contributes more to the light absorption of BrC in summer? What distinguishes the conclusion of your research from the previous studies mentioned?**

Response: Thank you for your questions. Indeed, higher solar radiation and temperatures in summer facilitate the secondary generation of BrC, with secondary BrC being more enriched in the WSOC. Previous studies have observed a significant contribution of secondary sources to WSBrC during summer (Du et al., 2014; Yan et al., 2017). Conversely, the WIOC tends to be more enriched with primary sources such as BB and coal combustion. As temperatures rise in summer, coal consumption typically declines. Consequently, during summer, $Abs_{365, WIOC}$ is predominantly contributed by BB. These points have been clarified in the revised manuscript

*The manuscript is revised as follows:*

**Lines 421 to 427:** *"In summer, the elevated solar radiation and temperatures can promote the secondary generation of BrC, with secondary BrC being more enriched in the WSOC. Previous studies have observed a significant contribution of secondary sources to WSBrC during summer (Du et al., 2014; Yan et al., 2017). However, the BrC within WIOC tends to be more enriched with primary sources such as BB and coal combustion (Figure 4c). As temperatures rise in summer, coal consumption typically declines. Consequently, during summer, $Abs_{365, WIOC}$ is predominantly contributed by*

*BB (Figure 4d)."*

**14) Reviewer's comment:**

**Lines 398-411: Was the correlation between biomass burning source contribution and the light absorption of WIOC discussed by the author? Was there a correlation similar to that observed with coal combustion?**

Response: Thanks for your questions. We conducted correlation analysis between the MAE365 of WIOC and the contribution of biomass burning. As shown in Figure R1, unlike the strong positive correlation observed for coal combustion ($r = 0.72$, $p < 0.01$), the contribution of biomass burning exhibited a negative correlation with the MAE365 of WIOC ($r = -0.34$, $p = 0.46$). Thus, the light-absorbing compounds derived from coal combustion have a stronger light-absorbing capacity than BB, and enhanced the overall MAE365 of WIOC. In the revised manuscript, we have added more discussion on this point to provide further insights into the contrasting contributions of coal combustion and biomass burning to the light-absorbing properties of WIOC.

[Figure]

Figure R1. The relationship between the $MAE_{365}$ of WIOC and relative contribution of biomass burning.

***The manuscript is revised as follows:***

**Lines 433 to 440:** *"Both BB and coal combustion are recognized as the sources*

*of BrC with strong light-absorbing capacity. We found that contribution of BB exhibited a negative correlation with $MAE_{365,WIOC}$ (r = -0.34, p = 0.46, Figure S3b), whereas a strong positive correlation was observed for coal combustion (r = 0.72, p < 0.01, Figure S3a). These suggest that the light-absorbing compounds derived from coal combustion have a stronger light-absorbing capacity than BB, and enhanced the overall $MAE_{365,WIOC}$."*

**15) Reviewer's comment:**

**Line 423: Figure 1c should be Figure 2c?**

Response: Corrected.

**16) Reviewer's comment:**

**Line 504: Figures 2b and c should be Figures 3b and c?**

Response: Corrected

**References:**

Du, Z.Y., He, K.B., Cheng, Y., Duan, F.K., Ma, Y.L., Liu, J.M., Zhang, X.L., Zheng, M., Weber, R. (2014), A yearlong study of water-soluble organic carbon in Beijing II: Light absorption properties, *Atmospheric Environment*, *89*, 235-241.https://doi.org/10.1016/j.atmosenv.2014.02.022

Liu, J., Bergin, M., Guo, H., King, L., Kotra, N., Edgerton, E., Weber, R.J. (2013), Size-resolved measurements of brown carbon in water and methanol extracts and estimates of their contribution to ambient fine-particle light absorption, *Atmospheric Chemistry and Physics*, *13*(24), 12389-12404.https://doi.org/10.5194/acp-13-12389-2013

Yan, C., Zheng, M., Bosch, C., Andersson, A., Desyaterik, Y., Sullivan, A.P., Collett, J.L., Zhao, B., Wang, S., He, K., Gustafsson, O. (2017), Important fossil source contribution to brown carbon in Beijing during winter, *Scientific Reports*, *7*.https://doi.org/10.1038/srep43182